# Exploiting Chain Rule and Bayes' Theorem to Compare Probability Distributions

**Huangjie Zheng**
Department of Statistics & Data Science
The University of Texas at Austin
Austin, TX 78712
`huangjie.zheng@utexas.edu`

**Mingyuan Zhou**
McCombs School of Business
The University of Texas at Austin
Austin, TX 78712
`mingyuan.zhou@mccombs.utexas.edu`

## Abstract

To measure the difference between two probability distributions, referred to as the source and target, respectively, we exploit both the chain rule and Bayes' theorem to construct conditional transport (CT), which is constituted by both a forward component and a backward one. The forward CT is the expected cost of moving a source data point to a target one, with their joint distribution defined by the product of the source probability density function (PDF) and a source-dependent conditional distribution, which is related to the target PDF via Bayes' theorem. The backward CT is defined by reversing the direction. The CT cost can be approximated by replacing the source and target PDFs with their discrete empirical distributions supported on mini-batches, making it amenable to implicit distributions and stochastic gradient descent-based optimization. When applied to train a generative model, CT is shown to strike a good balance between mode-covering and mode-seeking behaviors and strongly resist mode collapse. On a wide variety of benchmark datasets for generative modeling, substituting the default statistical distance of an existing generative adversarial network with CT is shown to consistently improve the performance. PyTorch code is provided.

## 1 Introduction

Measuring the difference between two probability distributions is a fundamental problem in statistics and machine learning [1–3]. A variety of statistical distances, such as the Kullback–Leibler (KL) divergence [4], Jensen–Shannon (JS) divergence [5], maximum mean discrepancy (MMD) [6], and Wasserstein distance [7], have been proposed to quantify the difference. They have been widely used for generative modeling with different mode covering/seeking behaviors [8–13]. The KL divergence, directly related to both maximum likelihood estimation and variational inference [14–16], requires the two probability distributions to share the same support and is often inapplicable if either is an implicit distribution whose probability density function (PDF) is unknown [17–20]. Variational auto-encoders (VAEs) [8], the KL divergence based deep generative models, are stable to train, but often exhibit mode-covering behaviors in its generated data, producing blurred images. The JS divergence is directly related to the min-max loss of a generative adversarial net (GAN) when the discriminator is optimal [9], while the Wasserstein-1 distance is directly related to the min-max loss of a Wasserstein GAN [11], whose critic is optimized under the 1-Lipschitz constraint. However, it is difficult to maintain a good balance between the updates of the generator and discriminator/critic, making (Wasserstein) GANs notoriously brittle to train. MMD [6] is an RKHS-based statistical distance behind MMD-GANs [10, 21, 22], which have also shown promising results in generative modeling when trained with a min-max loss. Different from VAEs, these GAN-based models often exhibit mode dropping and face the danger of mode collapse if not well tuned during the training.

35th Conference on Neural Information Processing Systems (NeurIPS 2021).

In this paper, we introduce conditional transport (CT) as a new method to quantify the difference between two probability distributions, which will be referred to as the source distribution $p_X(\boldsymbol{x})$ and target distribution $p_Y(\boldsymbol{y})$, respectively. The construction of CT is motivated by the following observation: the difference between $p_X(\boldsymbol{x})$ and $p_Y(\boldsymbol{y})$ can be reflected by the expected difference of two dependent random variables $\boldsymbol{x}$ and $\boldsymbol{y}$, whose joint distribution $\pi(\boldsymbol{x}, \boldsymbol{y})$ is constrained by both $p_X(\boldsymbol{x})$ and $p_Y(\boldsymbol{y})$ in a certain way. Denoting $c(\boldsymbol{x}, \boldsymbol{y}) \geq 0$ as a cost function to measure the difference between points $\boldsymbol{x}$ and $\boldsymbol{y}$, such as $c(\boldsymbol{x}, \boldsymbol{y}) = \|\boldsymbol{x} - \boldsymbol{y}\|_2^2$, the expected difference is expressed as $\mathbb{E}_{\pi(\boldsymbol{x}, \boldsymbol{y})}[c(\boldsymbol{x}, \boldsymbol{y})]$. A basic way to constrain $\pi(\boldsymbol{x}, \boldsymbol{y})$ with both $p_X(\boldsymbol{x})$ and $p_Y(\boldsymbol{y})$ is to let $\pi(\boldsymbol{x}, \boldsymbol{y}) = p_X(\boldsymbol{x})p_Y(\boldsymbol{y})$, which means drawing $\boldsymbol{x}$ and $\boldsymbol{y}$ independently from $p_X(\boldsymbol{x})$ and $p_Y(\boldsymbol{y})$, respectively; this expected difference $\mathbb{E}_{p_X(\boldsymbol{x})p_Y(\boldsymbol{y})}[c(\boldsymbol{x}, \boldsymbol{y})]$ is closely related to the energy distance [23]. Another constraining method is to require both $\int \pi(\boldsymbol{x}, \boldsymbol{y})d\boldsymbol{y} = p_X(\boldsymbol{x})$ and $\int \pi(\boldsymbol{x}, \boldsymbol{y})d\boldsymbol{x} = p_Y(\boldsymbol{y})$, under which $\min_\pi \{\mathbb{E}_{\pi(\boldsymbol{x}, \boldsymbol{y})}[c(\boldsymbol{x}, \boldsymbol{y})]\}$ becomes the Wasserstein distance [7, 24–26].

A key insight of this paper is that by exploiting the chain rule and Bayes' theorem, there exist two additional ways to constrain $\pi(\boldsymbol{x}, \boldsymbol{y})$ with both $p_X(\boldsymbol{x})$ and $p_Y(\boldsymbol{y})$: 1) A forward CT that can be viewed as moving the source to target distribution; 2) A backward CT that reverses the direction. Our intuition is that given a source (target) point, it is more likely to be moved to a target (source) point closer to it. More specifically, if the target distribution does not provide good coverage of the source density, then there will exist source data points that lie in low-density regions of the target, making the expected cost of the forward CT high. Therefore, we expect that minimizing the forward CT will encourage the target distribution to exhibit a *mode-covering* behavior with respect to (*w.r.t.*) the source PDF. Reversing the direction, we expect that minimizing the backward CT will encourage the target distribution to exhibit a *mode-seeking* behavior *w.r.t.* the source PDF. Minimizing the combination of both is expected to strike a good balance between these two distinct behaviors.

To demonstrate the use of CT, we apply it to train implicit (or explicit) distributions to model both 1D and 2D toy data, MNIST digits, and natural images. The implicit distribution is defined by a deep generative model (DGM) that is simple to sample from. We provide empirical evidence to show how to control the mode-covering versus mode-seeking behaviors by adjusting the ratio of the forward CT versus backward CT. To train a DGM for natural images, we focus on adapting existing GANs, with minimal changes to their settings except for substituting the statistical distances in their loss functions with CT. We leave tailoring the network architectures and settings to CT for future study. Modifying the loss functions of various existing DGMs with CT, our experiments show consistent improvements in not only quantitative performance and generation quality, but also learning stability. Our code is available at `https://github.com/JegZheng/CT-pytorch`.

## 2 Chain rule and Bayes' theorem based conditional transport

Exploiting the chain rule and Bayes' theorem, we can constrain $\pi(\boldsymbol{x}, \boldsymbol{y})$ with both $p_X(\boldsymbol{x})$ and $p_Y(\boldsymbol{y})$ in two different ways, leading to the forward CT and backward CT, respectively. To define the forward CT, we use the chain rule to factorize the joint distribution as

$$\pi(\boldsymbol{x}, \boldsymbol{y}) = p_X(\boldsymbol{x})\pi_Y(\boldsymbol{y} \,|\, \boldsymbol{x}),$$

where $\pi_Y(\boldsymbol{y} \,|\, \boldsymbol{x})$ is a conditional distribution of $\boldsymbol{y}$ given $\boldsymbol{x}$. This construction ensures $\int \pi(\boldsymbol{x}, \boldsymbol{y})d\boldsymbol{y} = p_X(\boldsymbol{x})$ but not $\int \pi(\boldsymbol{x}, \boldsymbol{y})d\boldsymbol{x} = p_Y(\boldsymbol{y})$. Denote $d_\phi(\boldsymbol{h}_1, \boldsymbol{h}_2) \in \mathbb{R}$ as a function parameterized by $\phi$, which measures the difference between two vectors $\boldsymbol{h}_1, \boldsymbol{h}_2 \in \mathbb{R}^H$ of dimension $H$. While allowing $\int \pi(\boldsymbol{x}, \boldsymbol{y})d\boldsymbol{x} \neq p_Y(\boldsymbol{y})$, to appropriately constraint $\pi(\boldsymbol{x}, \boldsymbol{y})$ by $p_Y(\boldsymbol{y})$, we treat $p_Y(\boldsymbol{y})$ as the prior distribution, view $e^{-d_\phi(\boldsymbol{x}, \boldsymbol{y})}$ as an unnormalized likelihood term, and follow Bayes' theorem to define

$$\pi_Y(\boldsymbol{y} \,|\, \boldsymbol{x}) = e^{-d_\phi(\boldsymbol{x}, \boldsymbol{y})}p_Y(\boldsymbol{y})/Q(\boldsymbol{x}), \quad Q(\boldsymbol{x}) := \int e^{-d_\phi(\boldsymbol{x}, \boldsymbol{y})}p_Y(\boldsymbol{y})\mathrm{d}\boldsymbol{y}, \tag{1}$$

where $Q(\boldsymbol{x})$ is a normalization term that ensures $\int \pi_Y(\boldsymbol{y} \,|\, \boldsymbol{x})d\boldsymbol{y} = 1$. We refer to $\pi_Y(\boldsymbol{y} \,|\, \boldsymbol{x})$ as the forward "navigator," which specifies how likely a given $\boldsymbol{x}$ will be mapped to a target point $\boldsymbol{y} \sim p_Y(\boldsymbol{y})$. We now define the cost of the forward CT as

$$\mathcal{C}(X \to Y) = \mathbb{E}_{\boldsymbol{x} \sim p_X(\boldsymbol{x})}\mathbb{E}_{\boldsymbol{y} \sim \pi_Y(\cdot \,|\, \boldsymbol{x})}[c(\boldsymbol{x}, \boldsymbol{y})]. \tag{2}$$

In the forward CT, we expect large $c(\boldsymbol{x}, \boldsymbol{y})$ to typically co-occur with small $\pi_Y(\boldsymbol{y} \,|\, \boldsymbol{x})$ as long as $p_Y(\boldsymbol{y})$ provides a good coverage of the density of $\boldsymbol{x}$. Thus minimizing the forward CT cost is expected to encourage $p_Y(\boldsymbol{y})$ to exhibit a mode-covering behavior *w.r.t.* $p_X(\boldsymbol{x})$. Such kind of behavior is also

expected when minimizing the forward KL divergence as $\text{KL}(p_X||p_Y) = \mathbb{E}_{\boldsymbol{x} \sim p_X}\left[\ln \frac{p_X(\boldsymbol{x})}{p_Y(\boldsymbol{x})}\right]$, which calls for $p_Y(\boldsymbol{x}) > 0$ whenever $p_X(\boldsymbol{x}) > 0$.

Reversing the direction, we construct the backward CT, where the joint is factorized as $\pi(\boldsymbol{x}, \boldsymbol{y}) = p_Y(\boldsymbol{y})\pi_X(\boldsymbol{x}\,|\,\boldsymbol{y})$ and the backward navigator is defined as

$$\pi_X(\boldsymbol{x}\,|\,\boldsymbol{y}) = e^{-d_\phi(\boldsymbol{x},\boldsymbol{y})}p_X(\boldsymbol{x})/Q(\boldsymbol{y}), \quad Q(\boldsymbol{y}) := \int e^{-d_\phi(\boldsymbol{x},\boldsymbol{y})}p_X(\boldsymbol{x})\mathrm{d}\boldsymbol{x}. \tag{3}$$

This ensures $\int \pi(\boldsymbol{x}, \boldsymbol{y})d\boldsymbol{x} = p_Y(\boldsymbol{y})$; while allowing $\int \pi(\boldsymbol{x}, \boldsymbol{y})d\boldsymbol{y} \neq p_X(\boldsymbol{x})$, it constrains $\pi(\boldsymbol{x}, \boldsymbol{y})$ by treating $p_X(\boldsymbol{x})$ as the prior to construct $\pi_X(\boldsymbol{x}\,|\,\boldsymbol{y})$. The backward CT cost is now defined as

$$\mathcal{C}(X \leftarrow Y) = \mathbb{E}_{\boldsymbol{y} \sim p_Y(\boldsymbol{y})}\mathbb{E}_{\boldsymbol{x} \sim \pi_X(\cdot\,|\,\boldsymbol{y})}[c(\boldsymbol{x}, \boldsymbol{y})]. \tag{4}$$

In the backward CT, we expect large $c(\boldsymbol{x}, \boldsymbol{y})$ to typically co-occur with small $\pi_X(\boldsymbol{x}\,|\,\boldsymbol{y})$ as long as $p_X(\boldsymbol{x})$ has good coverage of the density of $\boldsymbol{y}$. Thus minimizing the backward CT cost is expected to encourage $p_Y(\boldsymbol{y})$ to exhibit a mode-seeking behavior *w.r.t.* $p_X(\boldsymbol{x})$. Such kind of behavior is also expected when minimizing the reverse KL divergence as $\text{KL}(p_Y||p_X) = \mathbb{E}_{\boldsymbol{x} \sim p_Y}\left[\ln \frac{p_Y(\boldsymbol{x})}{p_X(\boldsymbol{x})}\right]$, which allows $p_Y(\boldsymbol{x}) = 0$ when $p_X(\boldsymbol{x}) > 0$ and it is fine for $p_Y$ to just fit some portion of $p_X$.

In comparison to the forward and revers KLs, the proposed forward and backward CT are more broadly applicable as they don't require $p_X$ and $p_Y$ to share the same distribution support and have analytic PDFs. For the cases where the KLs can be evaluated, we introduce

$$\text{D}(X, Y) = \text{KL}(p_X||p_Y) - \text{KL}(p_Y||p_X)$$

as a formal way to quantify the mode-seeking and mode-covering behavior of $p_Y$ w.r.t. $p_X$, with $\text{D}(X, Y) > 0$ implying mode seeking and with $D(X, Y) < 0$ implying mode covering.

Combining both the forward and backward CTs, we now define the CT cost as

$$\mathcal{C}_\rho(X, Y) := \rho\mathcal{C}(X \to Y) + (1 - \rho)\mathcal{C}(X \leftarrow Y), \tag{5}$$

where $\rho \in [0, 1]$ is a parameter that can be adjusted to encourage $p_Y(\boldsymbol{y})$ to exhibit *w.r.t.* $p_X(\boldsymbol{x})$ mode-seeking ($\rho = 0$), mode-covering ($\rho = 1$), or a balance of two distinct behaviors ($\rho \in (0, 1)$). By definition we have $\mathcal{C}_\rho(X, Y) \geq 0$, where the equality can be achieved when $p_X = p_Y$ and the navigator parameter $\phi$ is optimized such that $e^{-d_\phi(\boldsymbol{x},\boldsymbol{y})}$ is equal to one if and only if $\boldsymbol{x} = \boldsymbol{y}$ and zero otherwise. We also have $\mathcal{C}_{\rho=0.5}(X, Y) = \mathcal{C}_{\rho=0.5}(Y, X)$. We fix $\rho = 0.5$ unless specified otherwise.

## 2.1 Conjugacy based analytic conditional distributions

Estimating the forward and backward CTs involves $\pi_Y(\boldsymbol{y}\,|\,\boldsymbol{x})$ and $\pi_X(\boldsymbol{x}\,|\,\boldsymbol{y})$, respectively. Both conditional distributions, however, are generally intractable to evaluate and sample from, unless $p_X(\boldsymbol{x})$ and $p_Y(\boldsymbol{y})$ are conjugate priors for likelihoods proportional to $e^{-d(\boldsymbol{x},\boldsymbol{y})}$, *i.e.*, $\pi_X(\boldsymbol{x}\,|\,\boldsymbol{y})$ and $\pi_Y(\boldsymbol{y}\,|\,\boldsymbol{x})$ are in the same probability distribution family as $p_X(\boldsymbol{x})$ and $p_Y(\boldsymbol{y})$, respectively. For example, if $d(\boldsymbol{x}, \boldsymbol{y}) = \|\boldsymbol{x} - \boldsymbol{y}\|_2^2$ and both $p_X(\boldsymbol{x})$ and $p_Y(\boldsymbol{y})$ are multivariate normal distributions, then both $\pi_X(\boldsymbol{x}\,|\,\boldsymbol{y})$ and $\pi_Y(\boldsymbol{y}\,|\,\boldsymbol{x})$ will follow multivariate normal distributions.

To be more specific, we provide a univariate normal based example, with $x, y, \phi, \theta \in \mathbb{R}$ and

$$p_X(x) = \mathcal{N}(0, 1), \; p_Y(y) = \mathcal{N}(0, e^\theta), \; d_\phi(x, y) = (x - y)^2/(2e^\phi), \; c(x, y) = (x - y)^2. \tag{6}$$

Here we have $\text{D}(X, Y) = \text{KL}[\mathcal{N}(0, 1)||\mathcal{N}(0, e^\theta)] - \text{KL}[\mathcal{N}(0, e^\theta)||\mathcal{N}(0, 1)] = \theta - sinh(\theta)$, which is positive when $\theta < 0$, implying mode-seeking, and negative when $\theta > 0$, implying mode-covering. As shown in Appendix C, we have analytic forms of the forward and backward navigators as

$$\pi_Y(y\,|\,x) = \mathcal{N}(\sigma(\theta - \phi)x, \sigma(\theta - \phi)e^\phi), \quad \pi_X(x\,|\,y) = \mathcal{N}(\sigma(-\phi)y, \sigma(\phi)),$$

where $\sigma(a) = 1/(1 + e^{-a})$ denotes the sigmoid function, and forward and backward CT costs as

$$\mathcal{C}(X \to Y) = \sigma(\phi - \theta)(e^\theta + \sigma(\phi - \theta)), \quad \mathcal{C}(X \leftarrow Y) = \sigma(\phi)(1 + \sigma(\phi)e^\theta).$$

As a proof of concept, we illustrate the optimization under CT using the above example, for which $\theta = 0$ is the optimal solution that makes $p_X = p_Y$. Thus when applying gradient descent to minimize the CT cost $\mathcal{C}_{\rho=0.5}(X, Y)$, we expect the generator parameter $\theta \to 0$ with proper learning dynamic, as long as the learning of the navigator parameter $\phi$ is appropriately controlled. This is confirmed by

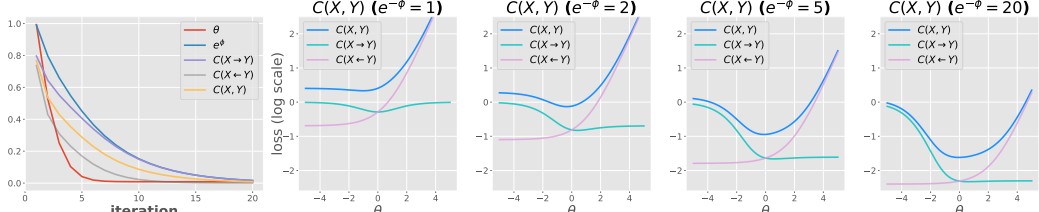

Figure 1: Illustration of minimizing the CT cost $\mathcal{C}_{\phi,\theta}(X, Y)$ between $\mathcal{N}(0, 1)$ and $\mathcal{N}(0, e^\theta)$. *Left*: Evolution of CT cost, its parameters, and forward and backward costs; *Right*: 4 CT cost curves against $\theta$ as $e^\phi$ is being optimized to a small value to jointly show the optimized $\phi$ provides better learning dynamic for the learning of $\theta$.

Fig. 1, which shows that as the navigator $\phi$ gets optimized by minimizing CT cost, it is more obvious that $\theta$ will minimize the CT cost at zero. This suggests that the navigator parameter $\phi$ mainly plays the role in assisting the learning of $\theta$. The right four subplots describe the log-scale curves of forward cost, backward cost and bi-directional CT costs *w.r.t.* $\theta$ as $\phi$ gets optimized to four different values. It is worth noting that the forward cost is minimized at $e^\theta > 1$, which implies a mode-covering behavior, and the backward cost is minimized at $e^\theta \to 0$, which implies a mode-seeking behavior, while the bi-directional cost is minimized at around the optimal solution $e^\theta = 1$; the forward CT cost exhibits a flattened curve on the right hand side of its minimum, adding to which the backward CT cost not only moves that minimum left, making it closer to $\theta = 0$, but also raises the whole curve on the right hand side, making the optimum of $\theta$ become easier to reach via gradient descent.

To apply CT in a general setting where the analytical forms of the distributions are unknown, there is no conjugacy, or we only have access to random samples from the distributions, below we show we can approximate the CT cost by replacing both $p_X(\boldsymbol{x})$ and $p_Y(\boldsymbol{y})$ with their corresponding discrete empirical distributions supported on mini-batches. Minimizing this approximate CT cost, amenable to mini-batch SGD based optimization, is found to be effective in driving the target (model) distribution $p_Y$ towards the source (data) distribution $p_X$, with the ability to control the mode-seeking and mode-covering behaviors of $p_Y$ *w.r.t.* $p_X$.

## 2.2 Approximate CT given empirical samples

Below we use generative modeling as an example to show how to apply the CT cost in a general setting that only requires access to random samples of both $\boldsymbol{x}$ and $\boldsymbol{y}$. Denote $\boldsymbol{x}$ as a data taking its value in $\mathbb{R}^V$. In practice, we observe a finite set $\mathcal{X} = \{\boldsymbol{x}_i\}_{i=1}^{|\mathcal{X}|}$, consisting of $|\mathcal{X}|$ data samples assumed to be *iid* drawn from $p_X(\boldsymbol{x})$. Given $\mathcal{X}$, the usual task is to learn a distribution to approximate $p_X(\boldsymbol{x})$, for which we consider a deep generative model (DGM) defined as $\boldsymbol{y} = G_{\boldsymbol{\theta}}(\boldsymbol{\epsilon})$, $\boldsymbol{\epsilon} \sim p(\boldsymbol{\epsilon})$, where $G_{\boldsymbol{\theta}}$ is a generator that transforms noise $\boldsymbol{\epsilon} \sim p(\boldsymbol{\epsilon})$ via a deep neural network parameterized by $\boldsymbol{\theta}$ to generate random sample $\boldsymbol{y} \in \mathbb{R}^V$. While the PDF of the generator, denoted as $p_Y(\boldsymbol{y}; \boldsymbol{\theta})$, is often intractable to evaluate, it is straightforward to draw $\boldsymbol{y} \sim p_Y(\boldsymbol{y}; \boldsymbol{\theta})$ with $G_{\boldsymbol{\theta}}$.

While knowing neither $p_X(\boldsymbol{x})$ nor $p_Y(\boldsymbol{y}; \boldsymbol{\theta})$, we can obtain discrete empirical distributions $p_{\hat{X}_N}$ and $p_{\hat{Y}_M}$ supported on mini-batches $\boldsymbol{x}_{1:N}$ and $\boldsymbol{y}_{1:M}$, as defined below, to guide the optimization of $G_{\boldsymbol{\theta}}$ in an iterative manner. With $N$ random observations sampled without replacement from $\mathcal{X}$, we define

$$p_{\hat{X}_N}(\boldsymbol{x}) = \tfrac{1}{N} \sum_{i=1}^{N} \delta(\boldsymbol{x} - \boldsymbol{x}_i), \quad \{\boldsymbol{x}_1, \ldots, \boldsymbol{x}_N\} \subseteq \mathcal{X} \tag{7}$$

as an empirical distribution for $\boldsymbol{x}$. Similarly, with $M$ random samples of the generator, we define

$$p_{\hat{Y}_M}(\boldsymbol{y}) = \tfrac{1}{M} \sum_{j=1}^{M} \delta(\boldsymbol{y} - \boldsymbol{y}_j), \; \boldsymbol{y}_j = G_{\boldsymbol{\theta}}(\boldsymbol{\epsilon}_j), \; \boldsymbol{\epsilon}_j \overset{iid}{\sim} p(\boldsymbol{\epsilon}) \; . \tag{8}$$

Substituting $p_Y(\boldsymbol{y}; \boldsymbol{\theta})$ in (2) with $p_{\hat{Y}_M}(\boldsymbol{y})$, the continuous forward navigator becomes a discrete one as

$$\hat{\pi}_Y(\boldsymbol{y} \,|\, \boldsymbol{x}) = \sum_{j=1}^{M} \hat{\pi}_M(\boldsymbol{y}_j \,|\, \boldsymbol{x}, \boldsymbol{\phi}) \delta_{\boldsymbol{y}_j}, \; \hat{\pi}_M(\boldsymbol{y}_j \,|\, \boldsymbol{x}, \boldsymbol{\phi}) := \frac{e^{-d_{\boldsymbol{\phi}}(\boldsymbol{x}, \boldsymbol{y}_j)}}{\sum_{j'=1}^{M} e^{-d_{\boldsymbol{\phi}}(\boldsymbol{x}, \boldsymbol{y}_{j'})}}. \tag{9}$$

Thus given $p_{\hat{Y}_M}$, the cost of a forward CT can be approximated as

$$\mathcal{C}_{\boldsymbol{\phi}, \boldsymbol{\theta}}(X \to \hat{Y}_M) = \mathbb{E}_{\boldsymbol{y}_{1:M} \overset{iid}{\sim} p_Y(\boldsymbol{y}; \boldsymbol{\theta})} \mathbb{E}_{\boldsymbol{x} \sim p_X(\boldsymbol{x})} \left[ \sum_{j=1}^{M} c(\boldsymbol{x}, \boldsymbol{y}_j) \hat{\pi}_M(\boldsymbol{y}_j \,|\, \boldsymbol{x}, \boldsymbol{\phi}) \right], \tag{10}$$

which can be interpreted as the expected cost of following the forward navigator to stochastically transport a random source point $\boldsymbol{x}$ to one of the $M$ randomly instantiated "anchors" of the target distribution. Similar to previous analysis, we expect this approximate forward CT to stay small as long as $p_Y(\boldsymbol{y}; \boldsymbol{\theta})$ exhibits a mode covering behavior *w.r.t.* $p_X(\boldsymbol{x})$.

Similarly, we can approximate the backward navigator and CT cost as

$$\hat{\pi}_X(\boldsymbol{x} \mid \boldsymbol{y}) = \sum_{i=1}^N \hat{\pi}_N(\boldsymbol{x}_i \mid \boldsymbol{y}, \boldsymbol{\phi}) \delta_{\boldsymbol{x}_i}, \quad \hat{\pi}_N(\boldsymbol{x}_i \mid \boldsymbol{y}, \boldsymbol{\phi}) := \frac{e^{-d_\phi(\boldsymbol{x}_i, \boldsymbol{y})}}{\sum_{i'=1}^N e^{-d_\phi(\boldsymbol{x}_{i'}, \boldsymbol{y})}},$$

$$\mathcal{C}_{\phi, \theta}(\hat{X}_N \leftarrow Y) = \mathbb{E}_{\boldsymbol{x}_{1:M} \overset{iid}{\sim} p_X(\boldsymbol{x})} \mathbb{E}_{\boldsymbol{y} \sim p_Y(\boldsymbol{y}; \boldsymbol{\theta})} \left[ \sum_{i=1}^N c(\boldsymbol{x}_i, \boldsymbol{y}) \hat{\pi}_N(\boldsymbol{x}_i \mid \boldsymbol{y}, \boldsymbol{\phi}) \right]. \qquad (11)$$

Similar to previous analysis, we expect this approximate backward CT to stay small as long as $p_Y(\boldsymbol{y}; \boldsymbol{\theta})$ exhibits a mode-seeking behavior *w.r.t.* $p_X(\boldsymbol{x})$.

Combining (10) and (11), we define the approximate CT cost as

$$\mathcal{C}_{\phi, \theta, \rho}(\hat{X}_N, \hat{Y}_M) = \rho \mathcal{C}_{\phi, \theta}(X \to \hat{Y}_M) + (1 - \rho) \mathcal{C}_{\phi, \theta}(\hat{X}_N \leftarrow Y), \qquad (12)$$

an unbiased sample estimate of which, given mini-batches $\boldsymbol{x}_{1:N}$ and $\boldsymbol{y}_{1:M}$, can be expressed as

$$\mathcal{L}_{\phi, \theta, \rho}(\boldsymbol{x}_{1:N}, \boldsymbol{y}_{1:M}) = \sum_{i=1}^N \sum_{j=1}^M c(\boldsymbol{x}_i, \boldsymbol{y}_j) \left( \frac{\rho}{N} \hat{\pi}_M(\boldsymbol{y}_j \mid \boldsymbol{x}_i, \boldsymbol{\phi}) + \frac{1-\rho}{M} \hat{\pi}_N(\boldsymbol{x}_i \mid \boldsymbol{y}_j, \boldsymbol{\phi}) \right)$$

$$= \sum_{i=1}^N \sum_{j=1}^M c(\boldsymbol{x}_i, \boldsymbol{y}_j) \left( \frac{\rho}{N} \frac{e^{-d_\phi(\boldsymbol{x}_i, \boldsymbol{y}_j)}}{\sum_{j'=1}^M e^{-d_\phi(\boldsymbol{x}_i, \boldsymbol{y}_{j'})}} + \frac{1-\rho}{M} \frac{e^{-d_\phi(\boldsymbol{x}_i, \boldsymbol{y}_j)}}{\sum_{i'=1}^N e^{-d_\phi(\boldsymbol{x}_{i'}, \boldsymbol{y}_j)}} \right). \qquad (13)$$

**Lemma 1.** *Approximate CT in* (12) *is asymptotic as* $\lim_{N, M \to \infty} \mathcal{C}_{\phi, \theta, \rho}(\hat{X}_N, \hat{Y}_M) = \mathcal{C}_{\phi, \theta, \rho}(X, Y)$.

### 2.3 Cooperatively-trained or adversarially-trained feature encoder

To apply CT for generative modeling of high-dimensional data, such as natural images, we need to define an appropriate cost function $c(\boldsymbol{x}, \boldsymbol{y})$ to measure the difference between two random points. A naive choice is some distance between their raw feature vectors, such as $c(\boldsymbol{x}, \boldsymbol{y}) = \|\boldsymbol{x} - \boldsymbol{y}\|_2^2$, which, however, is known to often poorly reflect the difference between high-dimensional data residing on low-dimensional manifolds. For this reason, with cosine similarity [27] as $\cos(\boldsymbol{h}_1, \boldsymbol{h}_2) := \frac{\boldsymbol{h}_1^T \boldsymbol{h}_2}{\sqrt{\boldsymbol{h}_1^T \boldsymbol{h}_1} \sqrt{\boldsymbol{h}_2^T \boldsymbol{h}_2}}$, we further introduce a feature encoder $\mathcal{T}_{\boldsymbol{\eta}}(\cdot)$, parameterized by $\boldsymbol{\eta}$, to help redefine the point-to-point cost and both navigators as

$$c_{\boldsymbol{\eta}}(\boldsymbol{x}, \boldsymbol{y}) = 1 - \cos(\mathcal{T}_{\boldsymbol{\eta}}(\boldsymbol{x}), \mathcal{T}_{\boldsymbol{\eta}}(\boldsymbol{y})), \quad d_\phi \left( \frac{\mathcal{T}_{\boldsymbol{\eta}}(\boldsymbol{x})}{\|\mathcal{T}_{\boldsymbol{\eta}}(\boldsymbol{x})\|}, \frac{\mathcal{T}_{\boldsymbol{\eta}}(\boldsymbol{y})}{\|\mathcal{T}_{\boldsymbol{\eta}}(\boldsymbol{y})\|} \right). \qquad (14)$$

To apply the CT cost to train a DGM, we find that the feature encoder $\mathcal{T}_{\boldsymbol{\eta}}(\cdot)$ can be learned in two different ways: 1) Cooperatively-trained: Training them cooperatively by alternating between two different losses: training the generator under a fixed $\mathcal{T}_{\boldsymbol{\eta}}(\cdot)$ with the CT loss, and training $\mathcal{T}_{\boldsymbol{\eta}}(\cdot)$ under a fixed generator with a different loss, such as the GAN discriminator loss, WGAN critic loss, and MMD-GAN [10] critic loss. 2) Adversarially-trained: Viewing the feature encoder as a critic and training it to maximize the CT cost, by not only inflating the point-to-point cost, but also distorting the feature space used to construct the forward and backward navigators' conditional distributions.

To be more specific, below we present the details for the adversarial way to train $\mathcal{T}_{\boldsymbol{\eta}}$. Given training data $\mathcal{X}$, to train the generator $G_{\boldsymbol{\theta}}$, forward navigator $\pi_\phi(\boldsymbol{y} \mid \boldsymbol{x})$, backward navigator $\pi_\phi(\boldsymbol{x} \mid \boldsymbol{y})$, and encoder $\mathcal{T}_{\boldsymbol{\eta}}$, we view the encoder as a critic and propose to solve a min-max problem as

$$\min_{\phi, \theta} \max_{\boldsymbol{\eta}} \mathbb{E}_{\boldsymbol{x}_{1:N} \subseteq \mathcal{X}, \, \boldsymbol{\epsilon}_{1:M} \overset{iid}{\sim} p(\boldsymbol{\epsilon})} [\mathcal{L}_{\phi, \theta, \rho, \boldsymbol{\eta}}(\boldsymbol{x}_{1:N}, \{G_{\boldsymbol{\theta}}(\boldsymbol{\epsilon}_j)\}_{j=1}^M)], \qquad (15)$$

where $\mathcal{L}_{\phi, \theta, \rho, \boldsymbol{\eta}}$ is defined the same as in (13), except that we replace $c(\boldsymbol{x}_i, \boldsymbol{y}_j)$ and $d_\phi(\cdot, \cdot)$ with their corresponding ones shown in (14) and use reparameterization in (8) to draw $\boldsymbol{y}_{1:M} := \{G_{\boldsymbol{\theta}}(\boldsymbol{\epsilon}_j)\}_{j=1}^M$. With SGD, we update $\phi$ and $\boldsymbol{\theta}$ using $\nabla_{\phi, \theta} \mathcal{L}_{\phi, \theta, \rho, \boldsymbol{\eta}}(\boldsymbol{x}_{1:N}, \{G_{\boldsymbol{\theta}}(\boldsymbol{\epsilon}_j)\}_{j=1}^M)$ and, if the feature encoder is adversarially-trained, update $\boldsymbol{\eta}$ using $-\nabla_{\boldsymbol{\eta}} \mathcal{L}_{\phi, \theta, \rho, \boldsymbol{\eta}}(\boldsymbol{x}_{1:N}, \{G_{\boldsymbol{\theta}}(\boldsymbol{\epsilon}_j)\}_{j=1}^M)$.

We find by experiments that both ways to learn the encoder work well, with the adversarial one generally providing better performance. It is worth noting that in (Wasserstein) GANs, while the adversarially-trained discriminator/critic plays a similar role as a feature encoder, the learning dynamics between the discriminator/critic and generator need to be carefully tuned to maintain training stability and prevent trivial solutions (*e.g.*, mode collapse). By contrast, the feature encoder of the CT cost based DGM can be stably trained in two different ways. Its update does not need to be well synchronized with the generator and can be stopped at any time of the training.

# 3   Related work

In practice, variational auto-encoders [8], the KL divergence based deep generative models, are stable to train, but often exhibit mode-covering behaviors and generate blurred images [28–32]. By contrast, both GANs and Wasserstein GANs can generate photo-realistic images, but they often suffer from stability and mode collapse issues, requiring the update of the discriminator/critic to be well synchronized with that of the generator. This paper introduces conditional transport (CT) as a new method to quantify the difference between two probability distributions. Deep generative models trained under CT not only allow the balance between mode-covering and mode-seeking behaviors to be adjusted, but also allow the encoder to be pretrained or frozen at any time during cooperative/adversarial training.

As the JS divergence requires the two distributions to have the same support, the Wasserstein distance is often considered as more appealing for generative modeling as it allows the two distributions to have non-overlapping support [24–26]. However, while GANs and Wasserstein GANs in theory are connected to the JS divergence and Wasserstein distance, respectively, several recent works show that they should not be naively understood as the minimizers of their corresponding statistical distances, and the role played by their min-max training dynamics should not be overlooked [33–35]. In particular, Fedus et al. [34] show that even when the gradient of the JS divergence does not exist and hence GANs are predicted to fail from the perspective of divergence minimization, the discriminator is able to provide useful learning signal. Stanczuk et al. [35] show that the dual form based Wasserstein GAN loss does not provide a meaningful approximation of the Wasserstein distance; while primal form based methods could better approximate the true Wasserstein distance, they in general clearly underperform Wasserstein GANs in terms of the generation quality for high-dimensional data, such as natural images, and require an inner loop to compute the transport plan for each mini-batch, leading to high computational cost [12, 35–38]. See previous works for discussions on the approximation error and gradient bias when estimating the Wasserstein distance with mini-batches [10, 23, 39, 40].

MMD-GAN [10, 21, 22] that calculates the MMD statistics in the latent space of a feature encoder is the most similar to the CT cost in terms of the actual loss function used for optimization. In particular, both the MMD-GAN loss and CT loss, given mini-batches $\boldsymbol{x}_{1:N}$ and $\boldsymbol{y}_{1:M}$, involve computing the differences of all $NM$ pairs $(\boldsymbol{x}_i, \boldsymbol{y}_j)$. Different from MMD-GAN, there is no need in CT to choose a kernel and tune its parameters. We provide below an ablation study to evaluate both 1) MMD generator + CT encoder and 2) MMD encoder + CT generator, which shows 1) performs on par with MMD, while 2) performs clearly better than MMD and on par with CT.

# 4   Experimental results

**Forward and backward analysis:**   To empirically verify our previous analysis of the mode covering (seeking) behavior of the forward (backward) CT, we train a DGM with (12) and show the corresponding interpolation weight from the forward CT cost to the backward one, which means $\text{CT}_\rho$ reduces from forward CT ($\rho = 1$), to the CT in (12) ($\rho \in (0, 1)$), and to backward CT ($\rho = 0$). We consider the squared Euclidean (i.e. $\mathcal{L}_2^2$) distance to define both cost $c(\boldsymbol{x}, \boldsymbol{y}) = \|\boldsymbol{x} - \boldsymbol{y}\|_2^2$ and $d_\phi(\boldsymbol{x}, \boldsymbol{y}) = \|\mathcal{T}_\phi(\boldsymbol{x}) - \mathcal{T}_\phi(\boldsymbol{y})\|_2^2$, where $\mathcal{T}_\phi$ denotes a neural network parameterized by $\phi$. We consider a 1D example of a bimodal Gaussian mixture $p_X(x) = \frac{1}{4}\mathcal{N}(x; -5, 1) + \frac{3}{4}\mathcal{N}(x; 2, 1)$ and a 2D example of 8-modal Gaussian mixture with equal component weight as in Gulrajani et al. [41]. We use an empirical sample set $\mathcal{X}$, consisting of $|\mathcal{X}| = 5,000$ samples from both 1D and 2D cases, and illustrate in Fig. 2 the KDE of 5000 generated samples $y_j = G_{\boldsymbol{\theta}}(\boldsymbol{\epsilon}_j)$ after 5000 training epochs. For the 1D case, we take 200 grids in $[-10, 10]$ to approximate the empirical distribution of $\hat{p}_X$ and $\hat{p}_Y$, and report the corresponding forward KL ($\text{KL}[\hat{p}_X || \hat{p}_Y]$), reverse KL ($\text{KL}[\hat{p}_Y || \hat{p}_X]$), and their difference $\text{D}(X, Y) = \text{KL}[\hat{p}_X || \hat{p}_Y] - \text{KL}[\hat{p}_X || \hat{p}_Y]$ below each corresponding sub-figure in Fig. 2.

Comparing the results of different $\rho$ in Fig. 2, it suggests that minimizing the forward CT cost only encourages the generator to exhibit mode-covering behaviors, while minimizing the backward CT cost only encourages mode-seeking behaviors. Combining both costs provides a user-controllable balance between mode covering and seeking, leading to satisfactory fitting performance, as shown in Columns 2-4. Note that for a fair comparison, we stop the fitting at the same iteration; in practice, we find if training with more iterations, both $\rho = 0.75$ and $\rho = 0.25$ can achieve comparable results as $\rho = 0.5$ in this example. Allowing the mode covering and seeking behaviors to be controlled by adjusting $\rho$ is an attractive property of $\text{CT}_\rho$.

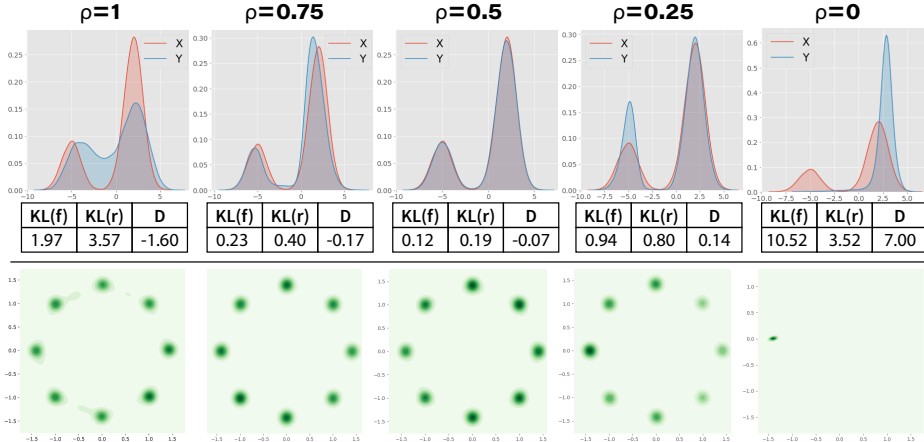

Figure 2: Forward and backward analysis: (*top*) Fitting 1D bi-modal Gaussian. Quantitative results of estimated forward KL (KL[$\hat{p}_X\|\hat{p}_Y$]), reverse KL (KL[$\hat{p}_Y\|\hat{p}_X$]), and the difference between the forward and reverse KL (D=KL[$\hat{p}_X\|\hat{p}_Y$]-KL[$\hat{p}_Y\|\hat{p}_X$]) are shown below each sub-figure. (*bottom*) 2D 8-Gaussian mixture by interpolating between the forward CT ($\rho = 1$) and backward CT ($\rho = 0$).

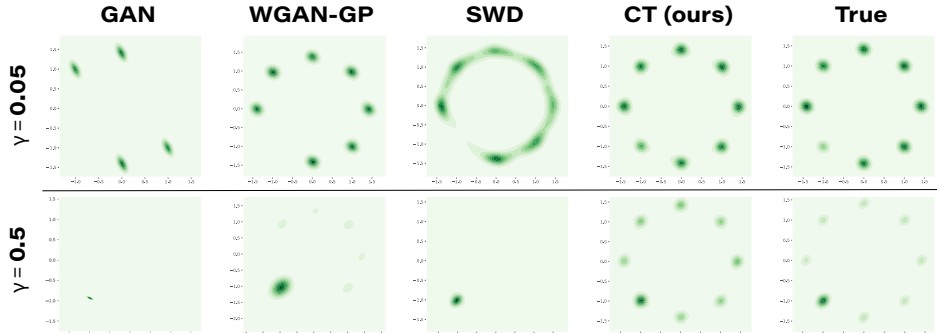

Figure 3: Experiments on the resistance to model collapse: Comparison of the generation quality on 8-Gaussian mixture data: one of the 8 modes has weight $\gamma$ and the rest modes have equal weight as $\frac{1-\gamma}{7}$.

**Resistance to mode collapse:** We continue to use a 8-Gaussian mixture to empirically evaluate how well a DGM resists mode collapse. Unlike the data in Fig. 2, where 8 modes are equally weighted, here the mode at the left lower corner is set to have weight $\gamma$, while the other modes are set to have the same weight of $\frac{1-\gamma}{7}$. We set $\mathcal{X}$ with 5000 samples and the mini-batch size as $N = 100$. When $\gamma$ is lowered to 0.05, its corresponding mode is shown to be missed by GAN, WGAN, and SWD-based DGM, while well kept by the CT-based DGM. As an explanation, GANs are known to be susceptible to mode collapse; WGAN and SWD-based DGMs are sensitive to the mini-batch size, as when $\gamma$ equals to a small value, the samples from this mode will appear in the mini-batches less frequently than those from any other mode, amplifying their missing mode problem. Similarly, when $\gamma$ is increased to 0.5, the other modes are likely to be missed by the baseline DGMs, while the CT-based DGM does not miss any modes. The resistance of CT to mode dropping can be attributed to its forward component's mode-covering property. The backward's mode-seeking property further helps distinguish the density of each mode component to avoid making components of equal weight.

**CT for 2D toy data and robustness in adversarial feature extraction:** To test CT with more general cases, we further conduct experiments on 4 representative 2D datasets for generative modeling evaluation [41]: 8-Gaussian mixture, Swiss Roll, Half Moons, and 25-Gaussian mixture. We apply the vanilla GAN [9] and Wasserstein GAN with gradient penalty (WGAN-GP) [41] as two representatives of min-max DGMs that require solving a min-max loss. We then apply the generators trained under the sliced Wasserstein distance (SWD) [42] and CT cost as two representatives of min-max-free DGMs. Moreover, we include CT with an adversarial feature encoder trained with (14) to test the robustness of adversary and compare with the baselines in solving the min-max loss.

On each 2D data, we train these DGMs as one would normally do during the first $5k$ epochs. We then only train the generator and freeze all the other learnable model parameters, which means we freeze the discriminator in GAN, critic in WGAN, the navigator parameter $\phi$ of the CT cost, and both

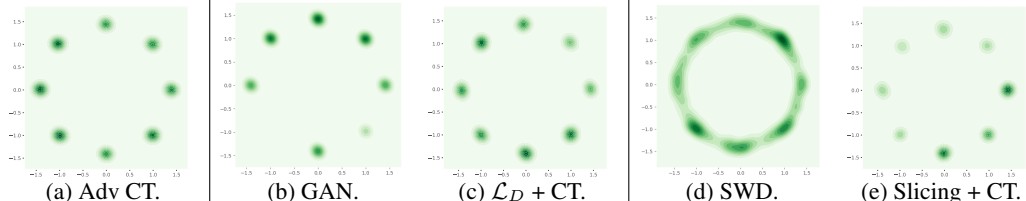

| (a) Adv CT. | (b) GAN. | (c) $\mathcal{L}_D$ + CT. | (d) SWD. | (e) Slicing + CT. |
|---|---|---|---|---|

Figure 4: Ablation of fitting results by minimizing CT in different spaces: (a) CT calculated with adversarially trained encoder. (b-c) GAN *vs.* CT with feature space cooperatively trained with discriminator loss. (d-f) Sliced Wasserstein distance and CT in the sliced space.

Table 1: FID comparison with different cooperative training on CIFAR-10 (lower FID is preferred).

| Critic space | FID ↓ |
|---|---|
| Discriminator | 29.7 |
| Slicing | 32.4 |
| Adversarial CT | **22.1** |

Table 2: FID Comparison with using MMD (Rational quadratic kernel/distance kernel) and CT loss in training critic/generator on CIFAR-10 (lower FID is preferred).

| MMD-rq | | Generator loss | | MMD-dist | | Generator loss | |
|---|---|---|---|---|---|---|---|
| | | MMD | CT | | | MMD | CT |
| Critic | MMD | 39.9 | 24.1 | Critic | MMD | 40.3 | **28.8** |
| loss | CT | 41.4 | **23.9** | loss | CT | 30.9 | 29.4 |

$(\phi, \eta)$ of CT with an adversarial feature encoder, for another $5k$ epochs. Figs. 7-10 in Appendix E.1 illustrate this training process on each dataset, where for both min-max baseline DGMs, the models collapse after the first $5k$ epochs, while the training for SWD remains stable and that for CT continues to improve. Compared to SWD, our method covers all data density modes and moves the generator much closer to the true data density. Notably, for CT with an adversarially trained feature encoder, although it has switched from solving a min-max loss to freezing the feature encoder after $5k$ epochs, the frozen feature encoder continues to guide the DGM to finish the training in the last $5k$ epochs, which shows the robustness of the CT cost.

**Ablation of cooperatively-trained and adversarially-trained CT:** As previous experiments show the adversarially-trained feature encoder could provide a valid feature space for CT cost, we further study the performance of the encoders cooperatively trained with other losses. Here we leverage, as two alternatives, the space of an encoder trained with the discriminator loss in GANs and the empirical Wasserstein distance in sliced 1D spaces [43]. We test these settings on both 8-Gaussian, as shown Fig. 4, and CIFAR-10 data, as shown in Table 1. It is confirmed these encoders are able to cooperatively work with CT, in general producing less appealing results with those trained by maximizing CT. From this view, although CT is able to provide guidance for the generators in the feature space learned with various options, maximizing CT is still preferred to ensure the efficiency. Moreover, as observed in Figs. 4b-4e, CT clearly improves the fitting with sliced Wasserstein distance. To explain why CT helps improve in the sliced space, we further provide a toy example in 1D to study the properties of CT and empirical Wasserstein distance in Appendix E.3.

**Ablation of MMD and CT:** As MMD also compares the pair-wise sample relations in a mini-batch, we study if MMD and CT can benefit each other. The feature space of MMD-GAN can be considered as $\mathcal{T}_{\eta} \circ k$, where $k$ is the rational quadratic or distance kernel in Bińkowski et al. [10]. Here we evaluate the combinations of MMD/CT as the generator/encoder criterion to train DGMs. On CIFAR-10, shown in Table 2, combining MMD and CT generally has improvement over MMD alone in FID. It is interesting to notice that for MMD-GAN, learning its generator with the CT cost shows more obvious improvement than learning its feature encoder with the CT cost. We speculate the estimation of MMD relies on a supremum of its witness function, which needs to be maximized *w.r.t* $\mathcal{T}_{\eta} \circ k$ and cannot be guaranteed by maximizing CT *w.r.t* $\mathcal{T}_{\eta}$. In the case of MMD-dist, using CT for witness function updates shows a more clear improvement, probably because CT has a similar form as MMD when using the distance kernel. From this view, CT and MMD are naturally able to be combined to compare the distributional difference with pair-wise sample relations. Different from MMD, CT does not involve the choice of kernel and its navigators assist to improve the comparison efficiency. Below we show on more image datasets, CT is compatible with many existing models, and achieve good results to show improvements on a variety of data with different scale.

**Adversarially-trained CT for natural images:** We conduct a variety of experiments on natural images to evaluate the performance and reveal the properties of DGMs optimized under the CT cost. We consider three widely-used image datasets, including CIFAR-10 [44], CelebA [45], and

Table 3: Results of CT with different deep generative models on CIFAR-10, CelebA and LSUN. Base model results are quoted from corresponding paper or github page.

| Method | Fréchet Inception Distance (FID ↓) | | | Inception Score (↑) |
| --- | --- | --- | --- | --- |
| | CIFAR-10 | CelebA | LSUN-bedroom | CIFAR-10 |
| DCGAN [49] | 30.2±0.9 | 52.5±2.2 | 61.7±2.9 | 6.2±0.1 |
| CT-DCGAN | **22.1±1.1** | **29.4±2.0** | **32.6±2.5** | **7.5±0.1** |
| SWG [42] | 33.7±1.5 | 21.9±2.0 | 67.9±2.7 | - |
| CT-SWG | **25.9± 0.9** | **18.8 ± 1.2** | **39.0 ± 2.1** | 6.9 ± 0.1 |
| MMD-GAN [10] | 39.9±0.3 | 20.6±0.3 | **32.0±0.3** | 6.5±0.1 |
| CT-MMD-GAN | **23.9 ± 0.4** | **13.8 ± 0.4** | 38.3 ± 0.3 | **7.4 ± 0.1** |
| SNGAN [50] | 21.5±1.3 | 21.7±1.5 | 31.1±2.1 | 8.2±0.1 |
| CT-SNGAN | **17.2±1.0** | **9.2±1.0** | **16.8±2.1** | **8.8±0.1** |
| StyleGAN2 [51] | 5.8 | 5.2 | **2.9** | 10.0 |
| CT-StyleGAN2 | **2.9 ± 0.5** | **4.0 ± 0.7** | 6.3 ± 0.2 | **10.1 ± 0.1** |

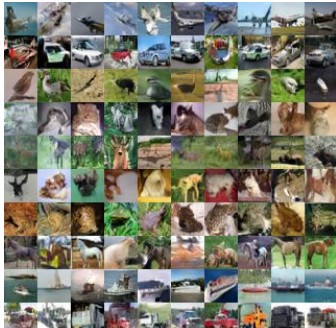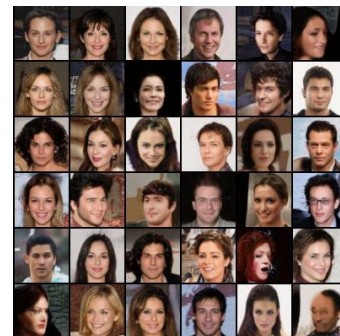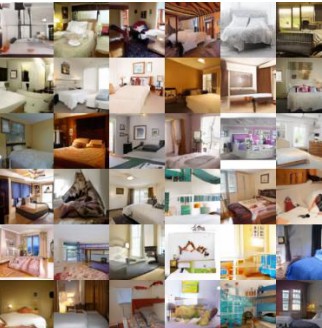

Figure 5: Generated samples of the deep generative model that adopts the backbone of SNGAN but is optimized with the CT cost on CIFAR-10, CelebA, and LSUN-Bedroom. See Appendix E for more results.

LSUN-bedroom [46] for general evaluation, as well as CelebA-HQ [47], FFHQ [48] for evaluation in high-resolution. We compare the results of DGMs optimized with the CT cost against DGMs trained with their original criterion including DCGAN [49], Sliced Wasserstein Generative model (SWG) [42], MMD-GAN [10], SNGAN [50], and StyleGAN2 [51]. For fair comparison, we leverage the best configurations reported in their corresponding paper or Github page. The detailed setups can be found in Appendix D. For evaluation metric, we consider the commonly used Fréchet inception distance (FID, lower is preferred) [52] on all datasets and Inception Score (IS, higher is preferred) [53] on CIFAR-10. Both FID and IS are calculated using a pre-trained inception model [54].

The summary of FID and IS on previously mentioned model is reported in Table 3. We observe that trained with CT cost, all the models have improvements with different margin in most cases, suggesting that CT is compatible with standard GANs, SWG, MMD-GANs, WGANs and generally helps improve generation quality, especially for data with richer modalities like CIFAR-10. CT is also compatible with advanced model architecture like StyleGAN2, confirming that a better feature space could make CT more efficient to guide the generator and produce better results.

The qualitative results shown in Fig. 5 are consistent with quantitative results in Table 3. To additionally show how CT works for more complex generation tasks, we show in Fig. 6 example higher-resolution images generated by CT-SNGAN on LSUN bedroom (128x128) and CelebA-HQ (256x256), as well as images generated by CT-StyleGAN2 on LSUN bedroom (256x256), FFHQ (256x256), and FFHQ (1024x1024).

**On the choice of $\rho$ for natural images:** In previous experiments, we fix $\rho = 0.5$ by default when we prefer neither mode-covering nor mode-seeking. We further tune $\rho$ as an additional ablation study on CIFAR-10 dataset with both the CT + DCGAN backbone and CT + SNGAN backbone to see its affects in terms of certain metrics, such as the FID score. The results

Table 4: FID of generation results on CIFAR-10, trained with different $\rho$.

| $\rho$ | 1 | 0.75 | 0.5 | 0.25 | 0 |
| --- | --- | --- | --- | --- | --- |
| CT-DCGAN | 25.1 | 22.1 | 22.1 | **21.4** | 72.1 |
| CT-SNGAN | 23.2 | 17.5 | **17.2** | **17.2** | 33.2 |

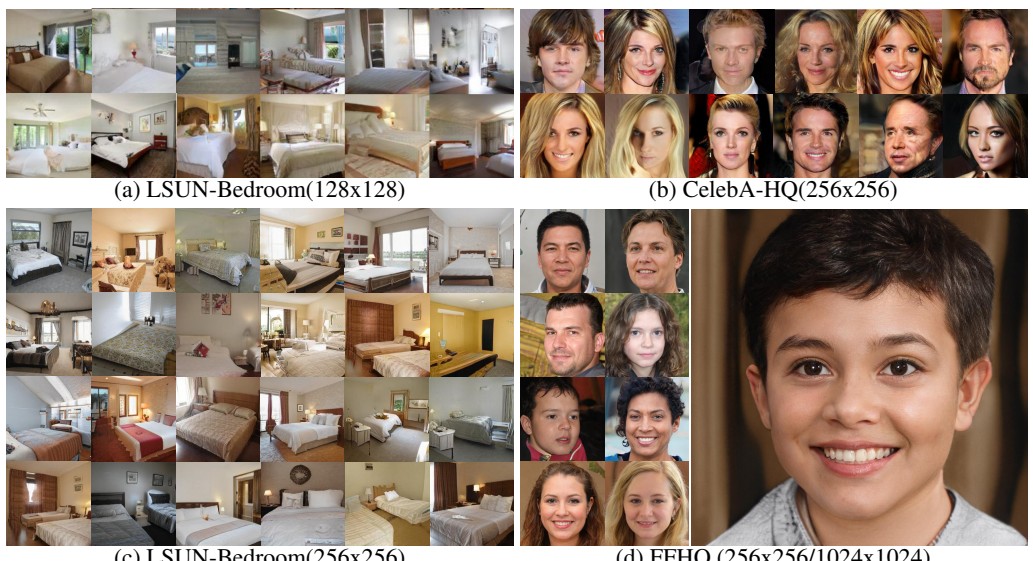

|               |               |
|:-------------:|:-------------:|
| (a) LSUN-Bedroom(128x128) | (b) CelebA-HQ(256x256) |
| (c) LSUN-Bedroom(256x256) | (d) FFHQ (256x256/1024x1024) |

Figure 6: Generation results in higher-resolution cases, with SNGAN and StyleGAN2 architecture. *Top*:LSUN-Bedroom (128x128) and CelebA-HQ (256x256), done with CT-SNGAN. *Bottom*: LSUN-Bedroom (256x256) and FFHQ (256x256/1024x1024), done with CT-StyleGAN2.

shown in Table 4 suggest that CT is not sensitive to the choice of $\rho$ as long as $0 < \rho < 1$, and the FID score could be further improved if we choose a smaller $\rho$ to bias towards mode-seeking.

## 5  Conclusion

We propose conditional transport (CT) as a new criterion to quantify the difference between two probability distributions, via the use of both forward and backward conditional distributions. The forward and backward expected cost are respectively with respect to a source-dependent and target-dependent conditional distribution defined via Bayes' theorem. The CT cost can be approximated with discrete samples and optimized with existing stochastic gradient descent-based methods. Moreover, the forward and backward CT possess mode-covering and mode-seeking properties, respectively. By combining them, CT nicely incorporates and balances these two properties, showing robustness in resisting mode collapse. On complex and high-dimensional data, CT is able to be calculated and stably guide the generative models in a valid feature space, which can be learned by adversarially maximizing CT or cooperatively deploying existing methods. On various benchmark datasets for deep generative modeling, we successfully train advanced models with CT. Our results consistently show improvement over the original ones, justifying the effectiveness of the proposed CT loss.

**Discussion:** Note CT brings consistent improvement to these DGMs by neither improving their network architectures nor gradient regularization. Thus it has great potential to work in conjunction with other state-of-the-art architectures or methods, such as BigGAN [55], self-attention GANs [56], partition-guided GANs [57], multimodal-DGMs [58], BigBiGAN [59], self-supervised learning [60], and data augmentation [61–63], which we leave for future study. As the paper is primarily focused on constructing and validating a new approach to quantify the difference between two probability distributions, we have focused on demonstrating the efficacy and interesting properties of the proposed CT on toy data and benchmark image data. We have focused on these previously mentioned models as the representatives in GAN, MMD-GAN, WGAN under CT, and we leave to future work using the CT to optimize more choices of DGMs, such as VAE-based models [8] and neural-SDE [64].

## Acknowledgments

The authors acknowledge the support of NSF IIS-1812699, the APX 2019 project sponsored by the Office of the Vice President for Research at The University of Texas at Austin, the support of a gift fund from ByteDance Inc., and the Texas Advanced Computing Center (TACC) for providing HPC resources that have contributed to the research results reported within this paper.

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
