# Exploiting Chain Rule and Bayes' Theorem to Compare Probability Distributions: Appendix

## A    Broader impact

This paper proposes to quantify the difference between two probability distributions with conditional transport, a bidirectional cost that we exploit to balance the mode seeking and covering behaviors of a generative model. The generative models trained with the proposed CT and datasets used in the experiments are classic in the area. Thus the capacities of these models are similar to existing ones, where we can see both positive and negative perspectives, depending on how the models are used. For example, good generative models can generate images for datasets that are expensive to collect, and be used to denoise and recover images. Meanwhile, they can also be misused to generate fake images for malicious purposes.

## B    Proof of Lemma 1

*Proof.* According to the strong law of large numbers, when $M \to \infty$, $\frac{1}{M} \sum_{j=1}^{M} e^{-d_\phi(\boldsymbol{x}, \boldsymbol{y}_j)}$, where $\boldsymbol{y}_j \overset{iid}{\sim} p_Y(\boldsymbol{y})$, converges almost surely to $\int e^{-d_\phi(\boldsymbol{x}, \boldsymbol{y})} p_Y(\boldsymbol{y}) d\boldsymbol{y}$ and $\frac{1}{M} \sum_{j=1}^{M} c(\boldsymbol{x}, \boldsymbol{y}_j) e^{-d_\phi(\boldsymbol{x}, \boldsymbol{y}_j)}$ converges almost surely to $\int c(\boldsymbol{x}, \boldsymbol{y}) e^{-d_\phi(\boldsymbol{x}, \boldsymbol{y})} p_Y(\boldsymbol{y}) d\boldsymbol{y}$. Thus when $M \to \infty$, the term $\sum_{j=1}^{M} c(\boldsymbol{x}, \boldsymbol{y}_j) \hat{\pi}_M(\boldsymbol{y}_j \mid \boldsymbol{x}, \boldsymbol{\phi})$ in (10) converges almost surely to $\frac{\int c(\boldsymbol{x}, \boldsymbol{y}) e^{-d_\phi(\boldsymbol{x}, \boldsymbol{y})} p_Y(\boldsymbol{y}) d\boldsymbol{y}}{\int e^{-d_\phi(\boldsymbol{x}, \boldsymbol{y})} p_Y(\boldsymbol{y}) d\boldsymbol{y}} = \int c(\boldsymbol{x}, \boldsymbol{y}) \pi_Y(\boldsymbol{y} \mid \boldsymbol{x}) d\boldsymbol{y}$. Therefore, $\mathcal{C}_{\phi, \theta}(X \to \hat{Y}_M)$ defined in (10) converges almost surely to the forward CT cost $\mathcal{C}_{\phi, \theta}(X \to Y)$ defined in (2) when $M \to \infty$. Similarly, we can show that $\mathcal{C}_{\phi, \theta}(\hat{X}_N \leftarrow Y)$ defined in (11) converges almost surely to the backward CT $\mathcal{C}_{\phi, \theta}(X \leftarrow Y)$ defined in (4) when $N \to \infty$.

$\square$

## C    Additional details for the univariate normal toy example shown in (6)

For the toy example specified in (6), exploiting the normal-normal conjugacy, we have an analytical conditional distribution for the forward navigator as

$$
\begin{aligned}
\pi_\phi(y \mid x) &\propto e^{-\frac{(x-y)^2}{2e^\phi}} \mathcal{N}(y; 0, e^\theta) \\
&\propto \mathcal{N}(x; y, e^\phi) \mathcal{N}(y; 0, e^\theta) \\
&= \mathcal{N}\left( \frac{e^\theta}{e^\theta + e^\phi} x, \frac{e^\phi e^\theta}{e^\theta + e^\phi} \right),
\end{aligned}
$$

and an analytical conditional distribution for the backward navigator as

$$
\begin{aligned}
\pi_\phi(x \mid y) &\propto e^{-\frac{(x-y)^2}{2e^\phi}} \mathcal{N}(x; 0, 1) \\
&\propto \mathcal{N}(y; x, e^\phi) \mathcal{N}(x; 0, 1) \\
&= \mathcal{N}\left( \frac{y}{1 + e^\phi}, \frac{e^\phi}{1 + e^\phi} \right).
\end{aligned}
$$

Plugging them into (2) and (4), respectively, and solving the expectations, we have

$$
\begin{aligned}
\mathcal{C}_{\phi, \theta}(\mu \to \nu) &= \mathbb{E}_{x \sim \mathcal{N}(0, 1)} \left[ \frac{e^\phi}{e^\theta + e^\phi} \left( e^\theta + \frac{e^\phi}{e^\theta + e^\phi} x^2 \right) \right] \\
&= \frac{e^\phi}{e^\theta + e^\phi} \left( e^\theta + \frac{e^\phi}{e^\theta + e^\phi} \right),
\end{aligned}
$$

$$\mathcal{C}_{\phi,\theta}(\mu \leftarrow \nu) = \mathbb{E}_{y \sim \mathcal{N}(0,e^\theta)} \left[ \frac{e^\phi}{1+e^\phi} \left( 1 + \frac{e^\phi}{1+e^\phi} y^2 \right) \right]$$
$$= \frac{e^\phi}{1+e^\phi} \left( 1 + \frac{e^\phi}{1+e^\phi} e^\theta \right).$$

## D  Experiment details

**Preparation of datasets**  We apply the commonly used training set of MNIST (50K gray-scale images, $28 \times 28$ pixels) [65], Stacked-MNIST (50K images, $28 \times 28$ with 3 channels pixels) [66], CIFAR-10 (50K color images, $32 \times 32$ pixels) [44], CelebA (about 203K color images, resized to $64 \times 64$ pixels) [45], and LSUN bedrooms (around 3 million color images, resized to $64 \times 64$ pixels) [46]. For MNIST, when calculate the inception score, we repeat the channel to convert each gray-scale image into a RGB format. For high-resolution generation, we use CelebA-HQ (30K images, resized to $256 \times 256$ pixels) [67] and FFHQ (70K images, with both original size $1024 \times 1024$ and resized size $256 \times 256$) [48]. All image pixels are normalized to range $[-1, 1]$.

**Experiment setups**  To avoid a large increase in model complexity, the navigator is parameterized as $d_\phi(\boldsymbol{x}, \boldsymbol{y}) := d_\phi((\boldsymbol{x} - \boldsymbol{y}) \circ (\boldsymbol{x} - \boldsymbol{y}))$, where $\circ$ denotes the Hadamard product, *i.e.*, the element-wise product. To be clear, we provide a Pytorch-like pseudo-code in Algorithm 1. For the toy datasets, we apply the network architectures presented in Table 5, where we set $H = 100$ for generator, navigator and feature encoder. For navigator, we set input dimension $V = 2$ and output dimension $d = 1$. If apply a feature encoder, we have $V = 2$, $d = 10$ for feature encoder and $V = 10$, $d = 1$ for navigator. The input dimension of generator is set as 50. The slopes of all leaky ReLU functions in the networks are set to 0.1 by default. We use the the Adam optimizer [68] with learning rate $\alpha = 2 \times 10^{-4}$ and $\beta_1 = 0.5$, $\beta_2 = 0.99$ for the parameters of the generator, and discriminator/critic. The learning rate of navigator is divided by 5. Typically, $5,000$ training epochs are sufficient. However, our experiments show that the DGM optimized with the CT cost can be stably trained at least over $10,000$ epochs (or possibly even more if allowed to running non-stop) regardless of whether the navigators are frozen or not after a certain number of iterations, where the GAN's discriminator usually diverges long before reaching that many training epochs even if we do not freeze it after a certain number of iterations.

For image experiments, to make the comparison fair, we strictly adopt the architecture of DCGAN [49][1], Sliced Wasserstein Generative model (SWG) [42][2], MMD-GAN [10][3], SNGAN [50][4], and StyleGAN2 [51][5], and follow their experiment setting: DCGAN and SWG apply CNN architecture on all datasets; MMD-GAN applies CNN on CIFAR-10 and ResNet architecture on other datasets; SN-GAN and StyleGAN2 apply their modified ResNet architecture. A summary of CNN and ResNet architecture is presented from Tables 7-12. To adapt the navigator, we apply the backbone of the discriminator in these GAN models as feature encoder and suppose the output dimension as $m$. The navigator is an MLP with architecture shown in Table 5 by setting $V = m$, $H = 512$, and $d = 1$. All models are able to be trained on a single GPU, Nvidia GTX 1080-TI/Nvidia RTX 3090 in our CIFAR-10, CelebA, LSUN-bedroom experiments. For high-resolution experiments, all experiments are done on 4 Tesla-V100-16G GPUs.

Table 5: Network architecture for toy datasets ($V$, $H$ and $d$ indicate the dimensionality).

| (a) Generator $G_\theta$ | (b) Navigator $d_\phi$ / Feature encoder $\mathcal{T}_\eta$ |
| --- | --- |
| $\boldsymbol{\epsilon} \in \mathbb{R}^{50} \sim \mathcal{N}(0,1)$ | $\boldsymbol{x} \in \mathbb{R}^V$ |
| $50 \to H$, dense, BN, lReLU | $V \to H$, dense, BN, lReLU |
| $H \to \lfloor \frac{H}{2} \rfloor$, dense, BN, lReLU | $H \to \lfloor \frac{H}{2} \rfloor$, dense, BN, lReLU |
| $\lfloor \frac{H}{2} \rfloor \to V$, dense, linear | $\lfloor \frac{H}{2} \rfloor \to d$, dense, linear |

---

[1]DCGAN architecture follows: https://github.com/pytorch/examples/tree/master/dcgan

[2]SWG architecture follows: https://github.com/ishansd/swg

[3]MMD-GAN architecture follows: https://github.com/mbinkowski/MMD-GAN

[4]SN-GAN architecture follows: https://github.com/pfnet-research/sngan_projection

[5]StyleGAN2 architecture follows: https://github.com/NVlabs/stylegan2. We use their config-f.

**Algorithm 1** PyTorch-like style pseudo-code of CT loss.

```
####################### Inputs #######################
# x: data B x C x W x H;
# y: generated samples B x C x W x H;
# netN: navigator network d -> 1
# netD: critic network C x W x H -> d
# rho: balance coefficient of forward-backward, default = 0.5

def ct_loss(x, y, netN, netD, rho):
    ####################### compute cost #######################
    f_x = netD(x) # feature of x: B x d
    f_y = netD(y) # feature of y: B x d
    cost = torch.norm(f_x[:,None] - f_y, dim=-1).pow(2) # pairwise cost: B x B

    ####################### compute transport map #######################
    mse_n = (f_x[:,None] - f_y).pow(2) # pairwise mse for navigator network: B x B x d
    d = netN(mse_n).squeeze().mul(-1) # navigator distance: B x B
    forward_map = torch.softmax(d, dim=1) # forward map is in y wise
    backward_map = torch.softmax(d, dim=0) # backward map is in x wise

    ####################### compute CT loss #######################
    # element-wise product of cost and transport map
    ct = rho * (cost * forward_map).sum(1).mean() + (1-rho) * (cost * backward_map).sum(0).mean()
    return ct
```

# E   Supplementary experiment results

## E.1   Results of 2D toy datasets and robustness in adversarial feature extraction

We visualize the results on the 8-Gaussian mixture toy dataset and other three commonly-used 2D toy datasets: Swiss-Roll, Half-Moon and 25-Gaussian mixture. As shown in Figs. 7-10, in the first 5k epochs, all DGMs are normally trained and the generative distributions are getting close to the true data distribution, while on 8-Gaussian and 25-Gaussian data, Vanilla GANs show mode missing behaviors. After 5k epochs, as the discriminator/navigator/feature encoder components in all DGMs are fixed, we can observe GAN and WGAN that solve min-max loss appear to collapse. This mode collapse issue of both GAN and WGAN-GP becomes more severe on the Swiss-Roll, Half-Moon, and 25-Gaussian datasets, since they rely on an optimized discriminator/critic to guide the generator. SWG relies on the slicing projection and is not affected, while its generated samples only cover the modes and ignore the correct density, indicating the effectiveness of slicing methods rely on the slicing [69]. The proposed CT cost show consistent good performance on the fitting of all these toy datasets, even after the navigator and the feature encoder are fixed after 5k epochs. This justifies our analysis about the robustness of CT cost.

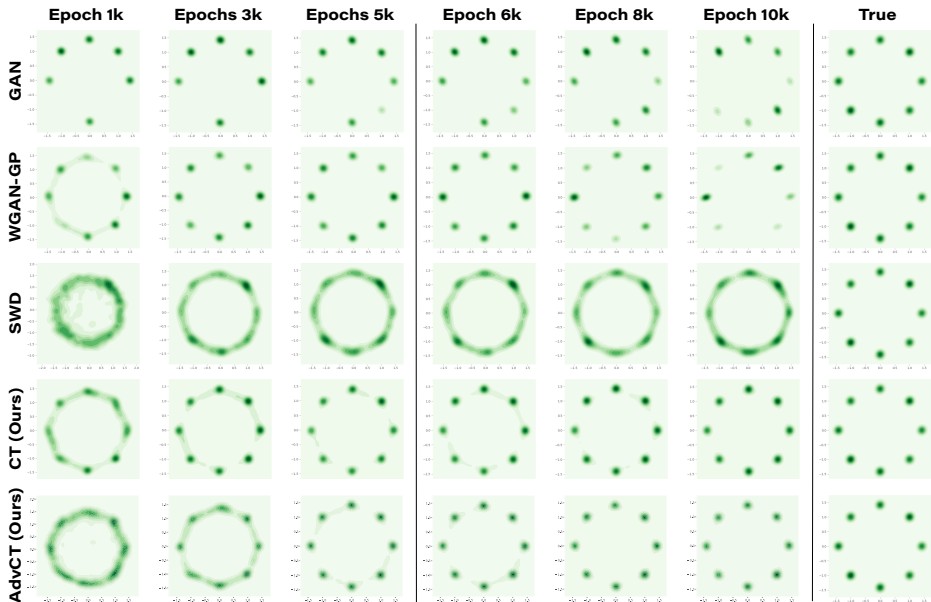

Figure 7: On a 8-Gaussian mixture data, comparison of generation quality and training stability between two min-max deep generative models (DGMs), including vallina GAN and Wasserstein GAN with gradient penalty (WGAN-GP), and two min-max-free DGMs, whose generators are trained under the sliced Wasserstein distance (SWD) and the proposed CT cost, respectively. The critics of GAN, WGAN-GP, the navigators of CT and the adversarially trained feature encoders of AdvCT are fixed after $5k$ training epochs. The last column shows the true data density.

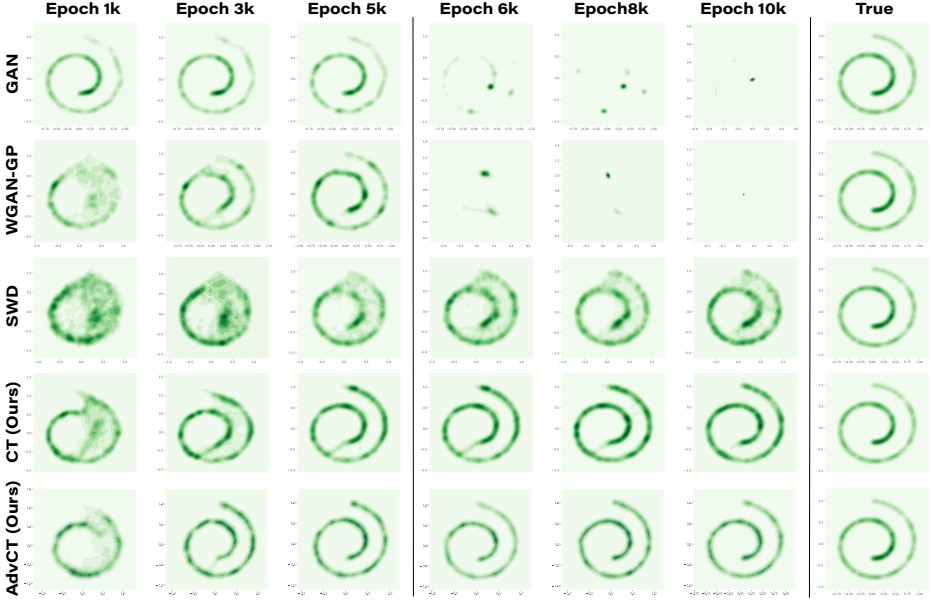

Figure 8: Analogous plot to Fig. 7 for the Swiss-Roll dataset.

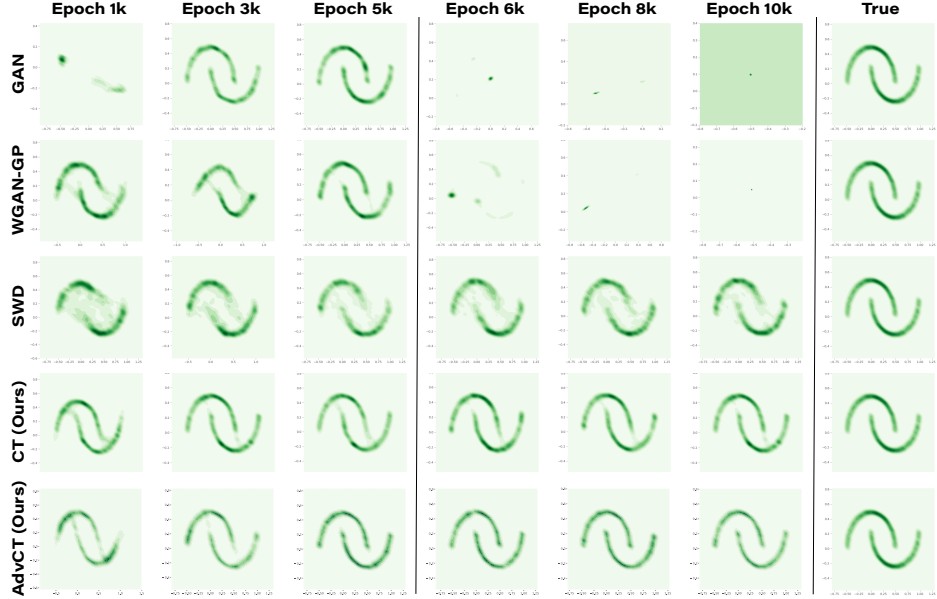

Figure 9: Analogous plot to Fig. 7 for the Half-Moon dataset.

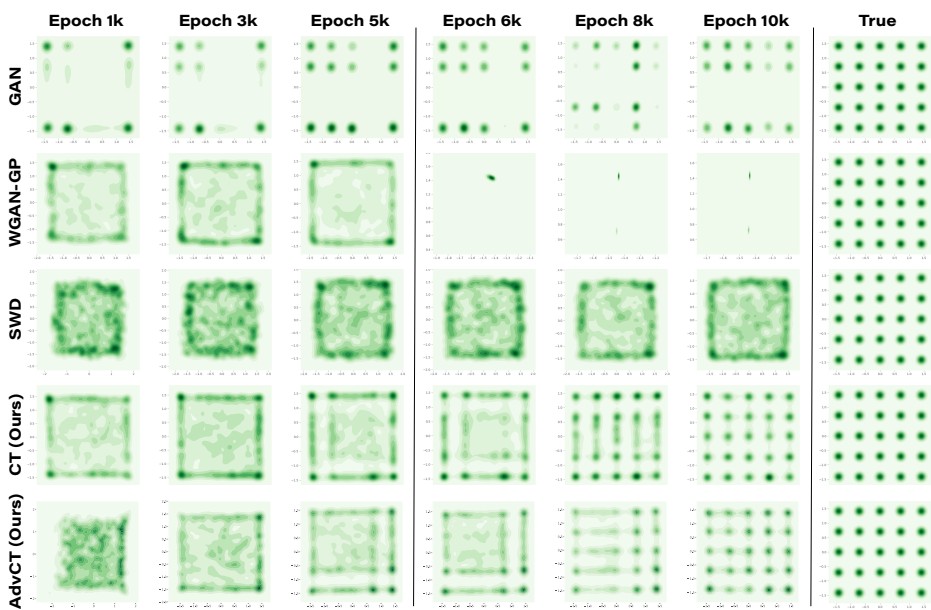

Figure 10: Analogous plot to Fig. 7 for the 25-Gaussian mixture dataset.

## E.2 Additional results of cooperative *vs.* adversarial encoder training

Here we provide additional results to the cooperative experiments, where we minimize CT in the feature encoder spaces trained by: 1) maximizing discriminator loss in GANs, 2) using random slicing projections, 3) maximizing MMD and 4) maximizing CT cost. Fig. 11 shows the results analogous to Fig. 4 on other three synthetic datasets: Swiss-Roll, Half-Moon and 25-Gaussian mixture. Fig. 12 provide qualitative results of Table 1 and Table 2.

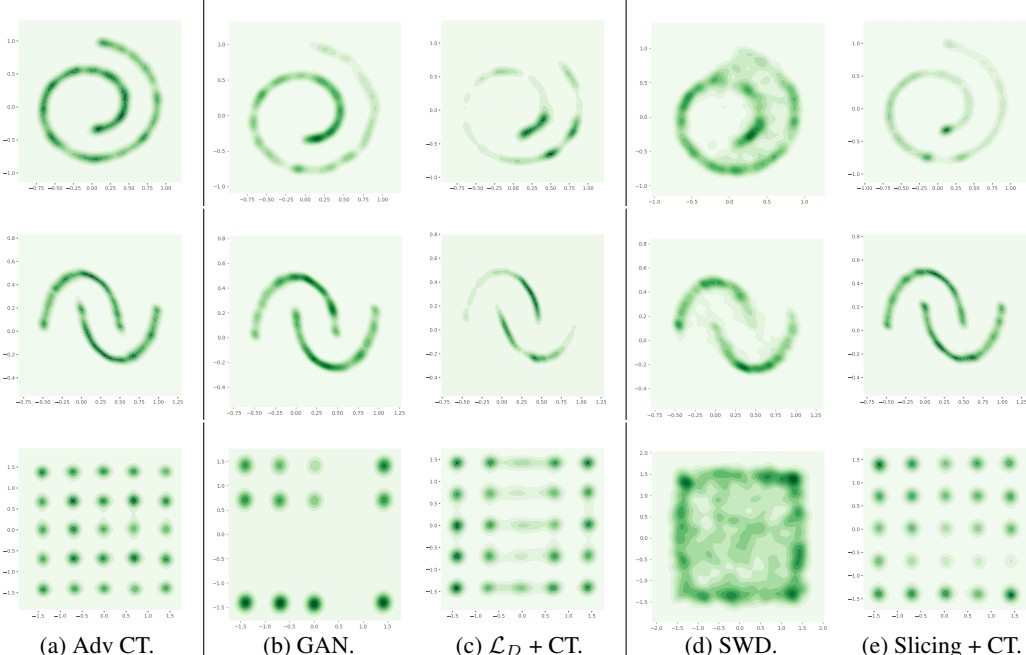

    (a) Adv CT.          (b) GAN.          (c) $\mathcal{L}_D$ + CT.          (d) SWD.          (e) Slicing + CT.

Figure 11: Analogous plot to Fig. 4 on Swiss roll, half-moon and 25 Gaussians datasets. Ablation of fitting results by minimizing CT in different spaces

## E.3 Empirical Wasserstein loss vs empirical CT

From Table 1, Fig. 4, and Fig. 11 we notice the proposed CT can improve the fitting with SWG [70] in the sliced 1D space. Considering SWG applies random slicing projections to project high-dimensional data to several 1D spaces, since the empirical Wasserstein distance has a close form in 1D case and can be calculated with ordered statistics, here we compare the empirical Wasserstein loss and empirical CT cost with a 1D toy experiments.

Let's consider the same 1D Gaussian mixture data used in Fig. 2, where the bimodal Gaussian mixture has a density form $p_X(x) = \frac{1}{4}\mathcal{N}(x; -5, 1) + \frac{3}{4}\mathcal{N}(x; 2, 1)$. We use an empirical sample set $\mathcal{X}$, consisting of $|\mathcal{X}| = 5,000$ samples, and train a generative model with the Wasserstein loss and CT cost estimated with these empirical data and generated samples. We vary the training mini-batch size from small to large. Fig. 13 shows the training curve *w.r.t.* each training epoch and the fitting results with mini-batch size 20, 200 and 5000. We can observe when the mini-batch size $N$ is as large as 5000, both Wasserstein and CT lead to a well-trained generator. However, as shown in the left and middle columns, when $N$ is getting much smaller, the generator trained with Wasserstein under-performs that trained with ACT, especially when the mini-batch size becomes as small as $N = 20$. While the Wasserstein distance $\mathcal{W}(X, Y)$ in theory can well guide the training of a generative model, the sample Wasserstein distance $\mathcal{W}(\hat{X}_N, \hat{Y}_N)$, whose optimal transport plan is locally re-computed for each mini-batch, could be sensitive to the mini-batch size $N$, which also explains why in practice the SWG are difficult to fit desired distribution. By contrast, CT shows better robustness across mini-batches, leading to a well-trained generator whose performance has low sensitivity to the mini-batch size.

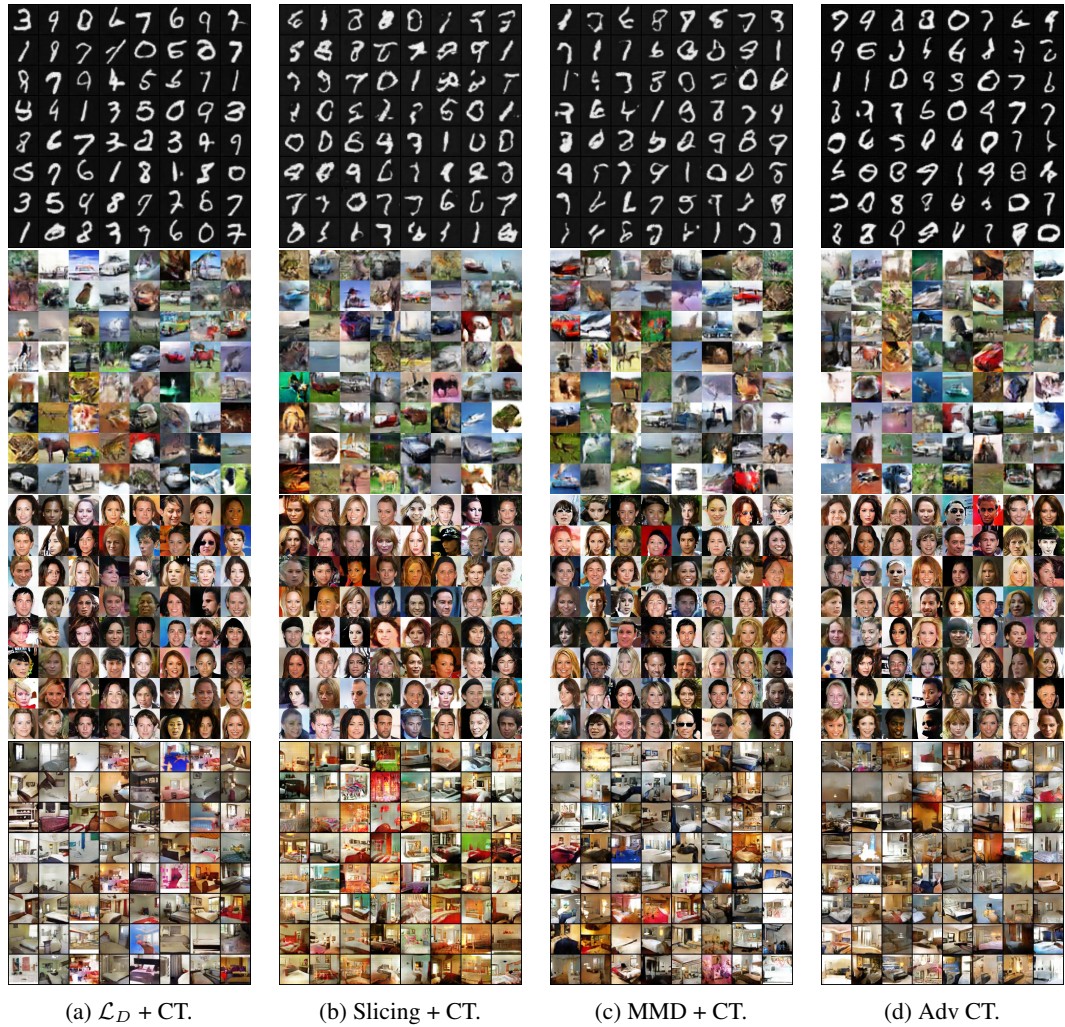

| (a) $\mathcal{L}_D$ + CT. | (b) Slicing + CT. | (c) MMD + CT. | (d) Adv CT. |

Figure 12: Analogous plot to Fig. 4 and Fig. 11 on image datasets. Ablation of fitting results by minimizing CT in different spaces.

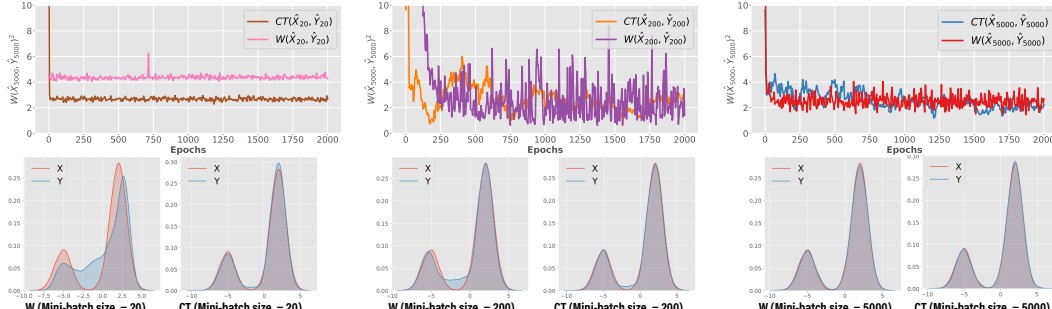

Figure 13: *Top*: Plot of the sample Wasserstein distance $W_2(\hat{X}_{5000}, \hat{Y}_{5000})^2$ against the number of training epochs, where the generator is trained with either $W_2(\hat{X}_N, \hat{Y}_N)^2$ or the CT cost between $\hat{X}_N$ and $\hat{Y}_N$, with the mini-batch size set as $N = 20$ (left), $N = 200$ (middle), or $N = 5000$ (right); one epoch consists of $5000/N$ SGD iterations. *Bottom*: The fitting results of different configurations, where the KDE curves of the data distribution and the generative one are marked in red and blue, respectively.

## E.4 Additional results on mode-covering/mode-seeking study

The mode covering and mode seeking behaviors discussed in Figs. 2 also exist in the real image case. For illustration, we use the Stacked-MNIST dataset [66] and fit CT in three configurations: normal, forward only, and backward only. DCGAN [49], VEEGAN [66], PacGAN [71], and PresGAN [72] are applied here as the baseline models to evaluate the mode-capturing capability.

Table 6: Assessing mode collapse on Stacked-MNIST. The true total number of modes is 1,000. DCGAN, VEEGAN, and CT (Backward only) all suffer from collapse. The other models capture nearly all the modes of the data distribution. Furthermore, the distribution of the labels predicted from the images produced by these models is closer to the data distribution, which shows lower KL scores.

| Method | Mode Captured | KL |
|---|---|---|
| DCGAN [49] | $392.0 \pm 7.376$ | $8.012 \pm 0.056$ |
| VEEGAN [66] | $761.8 \pm 5.741$ | $2.173 \pm 0.045$ |
| PacGAN [71] | $992.0 \pm 1.673$ | $0.277 \pm 0.005$ |
| PresGAN [72] | $999.4 \pm 0.80$ | $0.102 \pm 0.003$ |
| CT | $999.07 \pm 0.162$ | $0.181 \pm 0.003$ |
| CT (Foward only) | $999.18 \pm 0.9$ | $0.124 \pm 0.003$ |
| CT (Backward only) | $192 \pm 1.912$ | $9.166 \pm 0.06$ |

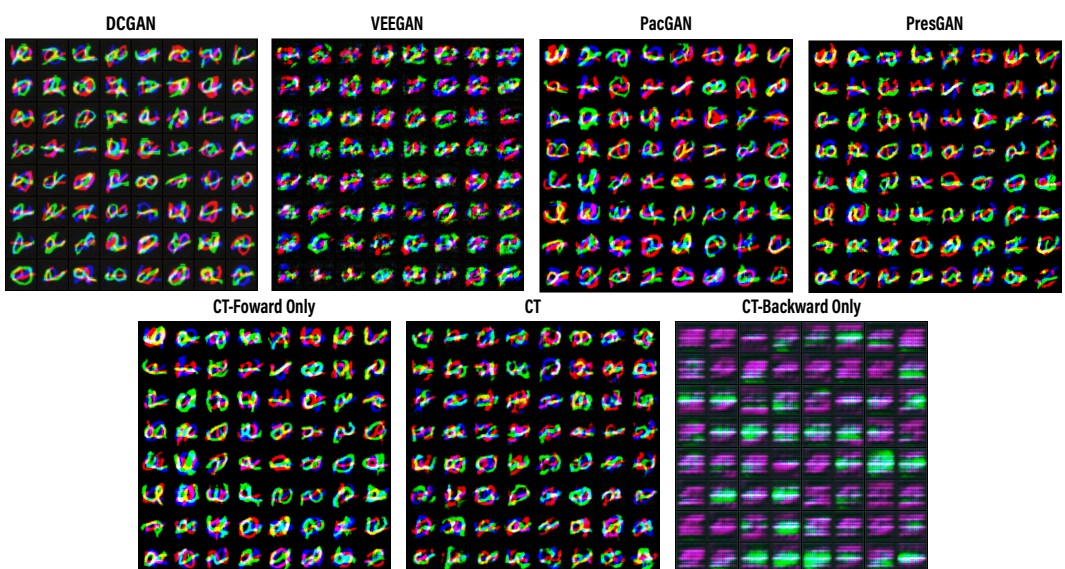

Figure 14: Visual results of the generated samples produced by DCGAN, VEEGAN, PacGAN, PresGAN, and ACT-DCGAN on the Stacked-MNIST dataset.

We calculate the captured mode number of each model, as well as the Kullback–Leibler (KL) divergence of the predicted label distributions between the generated samples and true data samples. For Stacked-MNIST data, there are 1000 modes in total. The results in Table 6 justify CT using only forward or using both forward and backward can almost capture all the modes, thus we do not suffer from the mode collapse problem. Using backward only can only encourages the mode seeking/dropping behavior. Fig. 14 provides the visual justification of this experiment, where the observations is consistent with those on toy datasets: if we only apply forward CT, the generator is encouraged to cover all the modes; if we only apply the backward CT for optimization, we can observe the mode seeking behavior of the generator.

## E.5 Additional results on image datasets

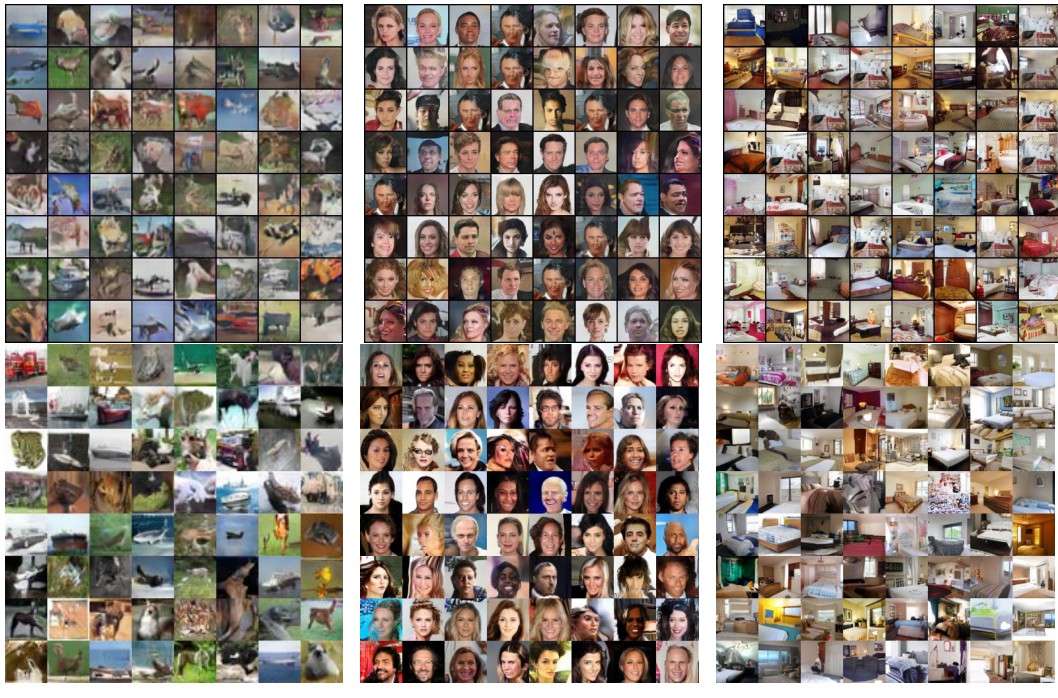

(a) CIFAR-10.        (b) CelebA.        (c) LSUN-Bedroom.

Figure 15: Analogous plot to Fig. 5, with additional generated samples. *Top*: samples generated with CNN backbone; *Bottom*: samples generated with ResNet backbone.

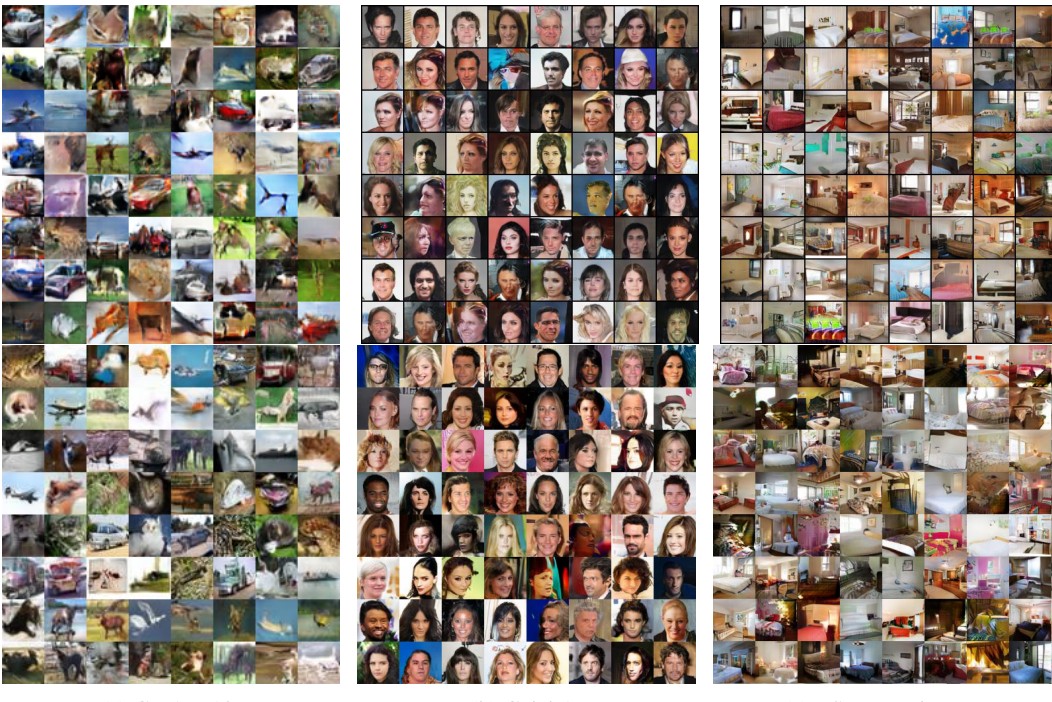

(a) CIFAR-10.        (b) CelebA.        (c) LSUN-Bedroom.

Figure 16: Analogous plot to Fig. 15.

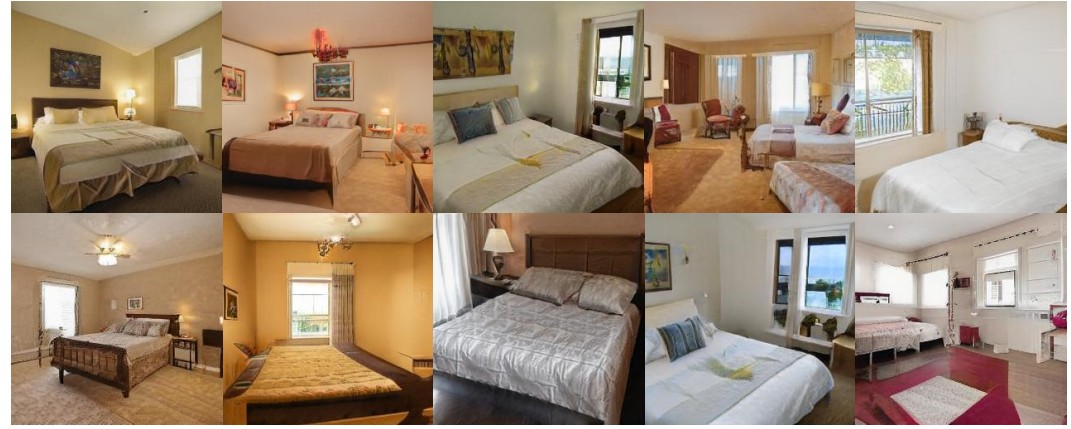

(a) LSUN-Bedroom (256x256).

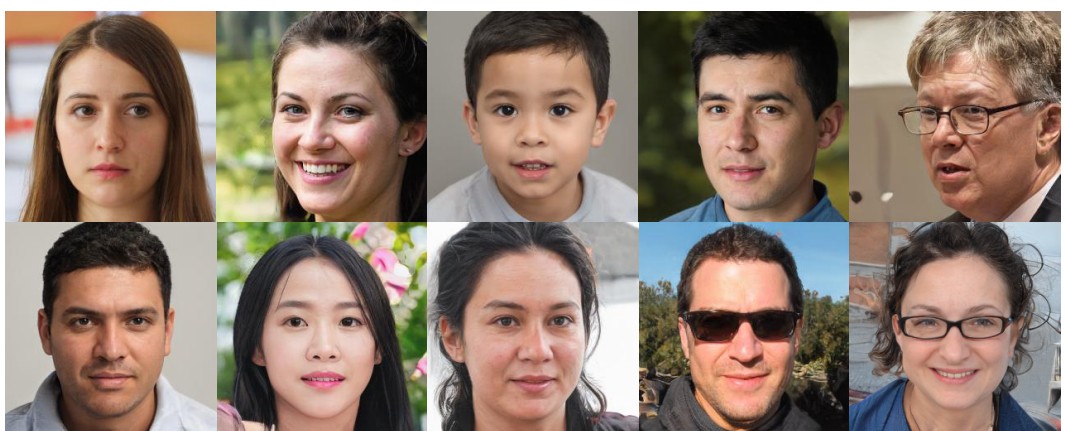

(b) FFHQ (256x256).

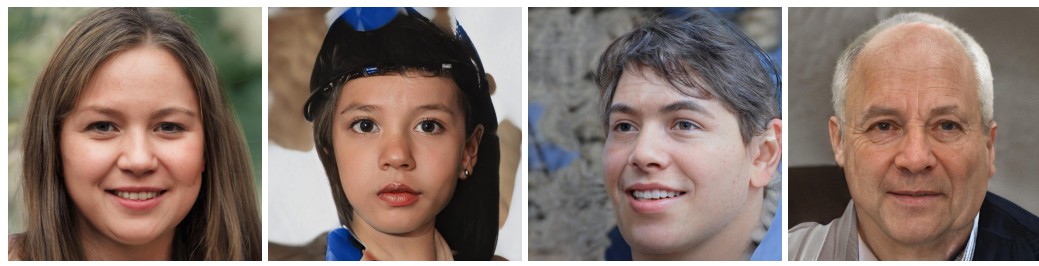

(c) FFHQ (1024x1024).

Figure 17: Analogous plot to Fig. 6: additional high-resolution samples.

# F Architecture summary

Table 7: DCGAN architecture for the CIFAR-10 dataset.

(a) Generator $G_{\boldsymbol{\theta}}$

| $\boldsymbol{\epsilon} \in \mathbb{R}^{128} \sim \mathcal{N}(0,1)$ |
| --- |
| $128 \to 4 \times 4 \times 512$, dense, linear |
| $4 \times 4$, stride=2 deconv. BN 256 ReLU |
| $4 \times 4$, stride=2 deconv. BN 128 ReLU |
| $4 \times 4$, stride=2 deconv. BN 64 ReLU |
| $3 \times 3$, stride=1 conv. 3 Tanh |

(b) Feature encoder $\mathcal{T}_{\boldsymbol{\eta}}$

| $\boldsymbol{x} \in [-1,1]^{32 \times 32 \times 3}$ |
| --- |
| $3 \times 3$, stride=1 conv 64 lReLU 
 $4 \times 4$, stride=2 conv 64 lReLU |
| $3 \times 3$, stride=1 conv 128 lReLU 
 $4 \times 4$, stride=2 conv 128 lReLU |
| $3 \times 3$, stride=1 conv 256 lReLU 
 $4 \times 4$, stride=2 conv 256 lReLU |
| $3 \times 3$, stride=1 conv. 512 lReLU |
| $h \times w \times 512 \to m$, dense, linear |

Table 8: DCGAN architecture for the CelebA and LSUN datasets.

(a) Generator $G_{\boldsymbol{\theta}}$

| $\boldsymbol{\epsilon} \in \mathbb{R}^{128} \sim \mathcal{N}(0,1)$ |
| --- |
| $128 \to 4 \times 4 \times 1024$, dense, linear |
| $4 \times 4$, stride=2 deconv. BN 512 ReLU |
| $4 \times 4$, stride=2 deconv. BN 256 ReLU |
| $4 \times 4$, stride=2 deconv. BN 128 ReLU |
| $4 \times 4$, stride=2 deconv. BN 64 ReLU |
| $3 \times 3$, stride=1 conv. 3 Tanh |

(b) Feature encoder $\mathcal{T}_{\boldsymbol{\eta}}$

| $\boldsymbol{x} \in [-1,1]^{64 \times 64 \times 3}$ |
| --- |
| $4 \times 4$, stride=2 conv 64 lReLU 
 $4 \times 4$, stride=2 conv BN 128 lReLU |
| $4 \times 4$, stride=2 conv BN 256 lReLU |
| $3 \times 3$, stride=1 conv BN 512 lReLU |
| $h \times w \times 512 \to m$, dense, linear, Normalize |

Table 9: ResNet architecture for the CIFAR-10 dataset.

(a) Generator $G_{\boldsymbol{\theta}}$

| $\boldsymbol{\epsilon} \in \mathbb{R}^{128} \sim \mathcal{N}(0,1)$ |
| --- |
| $128 \to 4 \times 4 \times 256$, dense, linear |
| ResBlock up 256 |
| ResBlock up 256 |
| ResBlock up 256 |
| BN, ReLU, $3 \times 3$ conv, 3 Tanh |

(b) Feature encoder $\mathcal{T}_{\boldsymbol{\eta}}$

| $\boldsymbol{x} \in [-1,1]^{32 \times 32 \times 3}$ |
| --- |
| ResBlock down 128 |
| ResBlock down 128 |
| ResBlock 128 |
| ResBlock 128 |
| ReLU |
| Global sum pooling |
| $h = 128 \to m$, dense, linear, Normalize |

Table 10: ResNet architecture for the CelebA and LSUN datasets.

(a) Generator $G_{\boldsymbol{\theta}}$

| $\boldsymbol{\epsilon} \in \mathbb{R}^{128} \sim \mathcal{N}(0,1)$ |
| --- |
| $128 \to 4 \times 4 \times 1024$, dense, linear |
| ResBlock up 512 |
| ResBlock up 256 |
| ResBlock up 128 |
| ResBlock up 64 |
| BN, ReLU, $3 \times 3$ conv, 3 Tanh |

(b) Feature encoder $\mathcal{T}_{\boldsymbol{\eta}}$

| $\boldsymbol{x} \in [-1,1]^{64 \times 64 \times 3}$ |
| --- |
| ResBlock down 128 |
| ResBlock down 256 |
| ResBlock down 512 |
| ResBlock down 1024 |
| ReLU Global sum pooling |
| $h = 1024 \to m$, dense, linear, Normalize |

Table 11: ResNet architecture for the LSUN-128 dataset.

(a) Generator $G_{\boldsymbol{\theta}}$

| $\boldsymbol{\epsilon} \in \mathbb{R}^{128} \sim \mathcal{N}(0,1)$ |
| --- |
| $128 \to 4 \times 4 \times 1024$, dense, linear |
| ResBlock up 1024 |
| ResBlock up 512 |
| ResBlock up 256 |
| ResBlock up 128 |
| ResBlock up 64 |
| BN, ReLU, $3 \times 3$ conv, 3 Tanh |

(b) Feature encoder $\mathcal{T}_{\boldsymbol{\eta}}$

| $\boldsymbol{x} \in [-1,1]^{128 \times 128 \times 3}$ |
| --- |
| ResBlock down 128 |
| ResBlock down 256 |
| ResBlock down 512 |
| ResBlock down 1024 |
| ResBlock 1024 |
| ReLU Global sum pooling |
| $h = 1024 \to m$, dense, linear, Normalize |

Table 12: ResNet architecture for the CelebA-HQ dataset.

(a) Generator $G_{\boldsymbol{\theta}}$

| $\boldsymbol{\epsilon} \in \mathbb{R}^{128} \sim \mathcal{N}(0,1)$ |
| --- |
| $128 \to 4 \times 4 \times 1024$, dense, linear |
| ResBlock up 1024 |
| ResBlock up 512 |
| ResBlock up 512 |
| ResBlock up 256 |
| ResBlock up 128 |
| ResBlock up 64 |
| BN, ReLU, $3 \times 3$ conv, 3 Tanh |

(b) Feature encoder $\mathcal{T}_{\boldsymbol{\eta}}$

| $\boldsymbol{x} \in [-1,1]^{256 \times 256 \times 3}$ |
| --- |
| ResBlock down 128 |
| ResBlock down 256 |
| ResBlock down 512 |
| ResBlock down 512 |
| ResBlock down 1024 |
| ResBlock 1024 |
| ReLU Global sum pooling |
| $h = 1024 \to m$, dense, linear, Normalize |