# OpenReview forum: "Exploiting Chain Rule and Bayes' Theorem to Compare Probability Distributions"
_NeurIPS.cc/2021/Conference — NeurIPS 2021 Poster_

### Official Review · Reviewer_cVJH · 2021-07-13

**Rating:** 7
**Confidence:** 4

**Summary:**

The paper defines conditional transport (CT), a new divergence between probability distributions (Equation 5), and reports experimental findings where this divergence serve as a good loss function for training generative models over natural images (Table 3, Figure 4 and so on). Among the tunable parameters of the divergence, there is one parameter that modulates between the extremes of mode seeking and mode covering. In real-data experiments, generative models fitted using CT as training loss have low Frechet inception distance (FID) and high inception score, which overall indicate high-quality generated images.

**Limitations And Societal Impact:**

The authors have adequately addressed the limitations of their work. As for potential negative societal impact, for the general field of generative modelling, there is the danger of creating fake material (fake news, etc) that spreads mis-information. Although the paper is primarily concerned with natural images, it is good to be aware of the general dangers.

**Main Review:**

Originality. CT uses ideas from optimal transport but possesses novel modifications. In the broadest characterization, Equation 15, the parameter rho aims to balance between mode seeking and mode covering behaviors. For the focus area of generative model training,  CT has additional parameters phi and eta. Eta parametrizes a neural network which transforms the source/target from the natural space into some feature space, while phi parametrizes another network that compares feature representations. CT, when used as training loss, very frequently improves generative modelling quality compared to previous choices of training loss, regardless of the neural network architecture. Each pair of methods in Table 3 correspond to the same architecture: the top rows are image quality metrics under existing training losses, while bottom rows are metrics for CT as training loss.

Quality. The quality of the generated images in Table 3/Figure 4,5 justify the use of phi and eta in the CT definition (although I have a question about the experimental setup that I will say in the Clarity section). The comparisons in Table 3 appear thorough, as the baselines range from very high-quality (like StyleGAN2) to low-quality (like DCGAN). Figure 1 is another justification for the navigator phi. In this synthetic situation, phi governs the curvature of the CT loss as a function of the generator parameters: suitable phi seemingly makes the CT loss more convex in the generator parameters, hence easier to optimize (I also have a question about the setup/result of this experiment). Figure 2 justifies the use of rho in the CT definition. As expected, the extreme setting of rho = 0 correspond to mode seeking behavior while rho = 1 corresponds to mode covering. By picking rho in between, there is a balance between the two.

Clarity. I have some questions about experimental setups and results. For Section 2.1, what does it mean for the navigator phi to be optimized? Up to this point, we have not talked about changing either rho or phi in definition of CT (Equation 5). Later on in section 2.3, the paper discusses  options like cooperative-trained versus adversarially-trained -- is this what it means in Section 2.1 too? If so, then as early as Section 2.1, it would be good to describe how phi is determined. Still in Section 2.1, for Figure 1, because the four panels on the right have different y-scaling, it's hard to see how the curvature of C(X,Y) change for different phi. Perhaps you can fix the same y-axis across the panels. Or plot the four curves on the same figure. For the "Adversarially-trained CT for natural images" paragraph in Section 4, what is the choice of rho? Perhaps you mention rho in the appendix, but since rho plays a big part in your methodology, and is mentioned in other experiments (like "Forward and backward analysis"), it would be good to know what rho is for these experiments as well. I have some comments about legibility. In general, the figure panels are too small, and it's difficult to see on a printed copy of the paper. Perhaps it's worth removing unnecessary panels, so that the panels chosen for display are easier to read. The panels in Figure 2 are very small and I can't make out what the axis labels/axis ticks say. The panels in Figure 3 are larger, but the axis ticks are not legible. For the panels in Figure 4, there are too many samples, each being too small.  My final comments are about word-choice/phrasing. Mini-max should be changed to min-max (minimax is another concept that is different).

Significance. Since CT serves as a good as-is training criterion without requiring specialized deep generative model architecture, other researchers are likely to use CT in place of existing training criteria in their research, hence this paper is an important set of results.

**Time Spent Reviewing:**

3

---

> ### Author Response · Authors · 2021-08-10
> **Response to Reviewer cVJH**
>
> Thank you for your valuable comments and suggestions. Below please find our point-by-point response.
>
> > *Clarification on Sections 2.1 and 2.3*
>
> To learn the value of $\phi$, we use the strategy in Equation (15), i.e., update $\phi$ by minimizing CT. The options of cooperative-trained versus adversarially-trained in section 2.3 are for the training of the feature encoder $\eta$, while Section 2.1 does not involve the updates of $\eta$.
> The right panel of Fig. 1 illustrates the optimization process, where $e^\phi$ is getting smaller, and we illustrate the corresponding CT cost in terms of $\theta$ when $e^\phi = 1$,   $e^\phi = 1/2$, $e^\phi = 1/5$  and $e^\phi = 1/20$. We will rescale the four panels to ensure that they share have same y-scaling.
>
> > *Clarification on the choice of $\rho$ in Section 4*
>
> Please see “Response to Two Common Questions” shown above for the clarification on the choice of $\rho$.
>
>
> >  *Legibility of figures and word-phrasing*
>
> For the 1-dimensional density panels in Fig. 2(a), x-axis indicates the coordinates and y represents the density; for 2-dimensional data panels in Fig. 2 and 3, the x-axis and y-axis represent the coordinates and the color indicates the corresponding density. We will carefully reorganize the figures and fix the mis-used words to make sure their meanings are clear to the readers. Thanks!

---

### Official Review · Reviewer_M4DG · 2021-07-16

**Rating:** 6
**Confidence:** 2

**Summary:**

This paper proposes a new distance measure between two probability distributions by exploiting the chain rule and Bayes' theorem. The new distance measure is obtained by minimizing out an additional parameter from the conditional transport (CT) cost. The CT costs is a convex combination of a forward component and a backward one. The forward CT is the expected cost of moving a source data point to a target one, with their joint distribution defined by the product of the source probability density function (PDF) and a surrogate source-dependent conditional distribution, which is related to the target PDF via Bayes' theorem. The backward CT is defined by reversing the direction. The CT cost can be approximated with discrete samples and optimized with existing gradient descent-based methods. The new measure nicely balances the mode-covering property of forward CT and mode-seeking property of backward CT, showing strong robustness against mode collapse. Experiments on a wide variety of benchmark datasets show that substituting the default statistical distance of an existing generative adversarial network with CT consistently improves the performance.

**Ethical Concerns:**

None.

**Limitations And Societal Impact:**

Yes.

**Main Review:**

The paper is very well written and a pleasure to read. The methods in Section 2 are clearly explained and technically sound. The review of related work is informative. PyTorch code is provided to reproduce the experimental results.

Minor comments:
Lines 109-110: It would be useful to briefly explain the concepts "mode-covering" and "mode-seeking".
Eq. (7): Please define the symbol $\delta$. Does $\delta_{x_i}$ denote the Dirac delta function? It should be written as $\delta_{x_i}(x)$ or $\delta(x-x_i)$.
Line 135: "Substituting .... in (2)" should be "Substituting .... in (1)".


**Time Spent Reviewing:**

4

---

> ### Author Response · Authors · 2021-08-10
> **Response to Reviewer M4DG**
>
> Thank you for your suggestions to help us enhance the paper quality, we will revise our paper accordingly:
>
> > *Lines 109-110: It would be useful to briefly explain the concepts "mode-covering" and "mode-seeking".*
>
> Following your suggestion, in "Response to Two Common Questions" shown above, we have added explanations on both concepts in a general setting.
>
>
> For this specific toy example, since the true data distribution is $\mathcal N(0,1)$ and the generator is $\mathcal N(0,e^{\theta}) $, we will observe the generator to exhibit mode-covering behavior when $e^{\theta}>1$, which is found to be the case when only the forward CT cost is being optimized, and mode-seeking behavior when $e^{\theta}<1$, which is found to the case when only the backward CT cost is being optimized.
>
> > *Clarify the definition of Eq. (7)*
>
> Following your suggestion, we will revise it as $p_{\hat X_N}(x) = \frac{1}{N} \sum_{i=1}^N \delta(x-x_i)$
>
> > *L. 135: "Substituting .... in (2)"  -> "Substituting .... in (1)"*
>
> We will make the correction.

---

> ### Author Response · Authors · 2021-08-29
> **Thank you for your support**
>
> Dear Reviewer M4DG,
>
> We appreciate your positive rating. We will carefully address your comments, as described in your response shown below. Please don't hesitate to let us know if you have any further comments and suggestions.

---

### Official Review · Reviewer_aP1u · 2021-07-20

**Rating:** 6
**Confidence:** 2

**Summary:**

This paper presents a new cost function to measure the difference between two probability distributions. The proposed cost function called the conditional transport (CT) is a mixture of two cost functions. The two cost functions of the mixture are derived by factorizing the joint probability density function in two reverse manners. A sample analogue of the CT is given, and it is shown that this sample analogue converges to its population version as the number of samples tends to infinity. Experiments are given to assess the performance of the CT and compare the performance of CT-based methods with that of existing methods.

**Ethical Concerns:**

There do not seem to be ethical issues with this paper.

**Limitations And Societal Impact:**

I do not find much discussion about the the limitations of the proposed method. I am a little worried about whether the estimation of $\rho$ of the cost function (12) is computationally expensive or not. (I ask this because the selection of tuning parameters of a loss function requires a lot of computation in general.)

I do not find any negative societal impact of the work.

**Main Review:**

Originality: The presented theory seems new. The proposed cost function, CT, comes from an interesting idea to factorize the joint probability density function through the chain rule and Bayes' theorem in two reverse manners. I am impressed with a comprehensive experimental study to demonstrate the performance of the CT and compare the performance of methods based on the CT with that of existing methods. Related works are adequately cited.

Quality: The submission seems technically sound in general. The claims are well supported by experimental results. Compared with great effort on the experiments, relatively little results are given to the theory of the CT. But Lemma 1 proves the convergence of the sample version of the proposed test to its expected version. The advantages of the proposed method are demonstrated through experiments. I do not find much discussion about the weaknesses of the proposed method in the paper.

Clarity: The paper is clearly written in general and well organized. The following are some minor comments on the presentation of the paper:

(i) p.2, l.71: The definition of $\mathbb{R}^H$ should be given.

(ii) p.3, l.86: $p_{\boldsymbol{\theta}}(\boldsymbol{y}) \ $  $\Longrightarrow$ $ \ p_{Y}(\boldsymbol{y})$?

(iii) p.4, l.127: What does "DGM" stand for?

(iv) p.4, equation (7): The definition of $\delta_{ \boldsymbol{x_i} } $ should be given. As it stands, the left-hand side of the equation says $p_{\hat{X}_N}$ is a function of $\boldsymbol{x}$, but the right-hand side of the equation does not involve $\boldsymbol{x}$.

Significance: The thorough experiments given in the paper suggest that the proposed method is useful in practice. Comparatively little work have been done on the theory of the proposed method. Although Lemma 1 is helpful, it would be better to have more theoretical results about the CT. For example, I think the selection of the weight $\rho$ of the cost function (12) would be a possible topic to discuss. If the true distribution is unknown, is it possible to select $\rho$ efficiently in practice?


**Time Spent Reviewing:**

5

---

> ### Author Response · Authors · 2021-08-10
> **Response to Reviewer aP1u**
>
> Thank you for your detailed comments to help us improve the presentation of this paper. Below we respond to your main concerns.
>
> > *Q1. More theoretical analysis*:
>
> Following your comments, in “Response to Two Common Questions” shown above, we have included additional analysis of the CT cost and formalized the definitions of the mode-covering and model-seeking behaviors.
>
> > *Q2. Limitations and selection of $\rho$*:
>
> We’d like to clarify how to select $\rho$ is not a limitation of the proposed CT, this is because we can simply set $\rho=0.5$, as used in all our experiments unless specified otherwise, to strike a good balance between the mode-covering and mode-seeking behaviors. With that said, we agree tuning $\rho$ for a specific dataset or downstream task could further improve the performance, which is out of the scope of this paper. Please see “Response to Two Common Questions” for more detailed discussions.
>
> One limitation we’d like to point out is that the proposed CT involves an extra parametric module $\phi$. In our experiments, we lighten this module as much as we can to a single MLP. According to our observations, increasing the parameter scale of $\phi$ could make $\pi_\phi$ more flexible and provide better results. How to better parameterize $\phi$ without clearly increasing the computational cost is left as our future work.
>
> > *Q3. Typos*:
>
> We will revise these points in our paper:
>
> L. 71: $\mathbb{R}^H$, where H is the dimensionality of the vector
>
> L. 86: $p_θ(y)$ -> $p_Y(y)$
>
> L. 127: DGM -> Deep Generative Model (DGM)
>
> Eqn. (7): $p_{\hat X_N}(x) = \frac{1}{N} \sum_{i=1}^N \delta(x-x_i)$

---

> > ### Comment · Reviewer_aP1u · 2021-08-29
> > **On the responses from authors**
> >
> > I would like to thank the authors for their detailed responses.
> >
> > The authors carefully responded to the comments I had made on the submitted paper. After reading their responses, my concern about the selection of the tuning parameter $\rho$ has been solved. It seems reasonable to select the value of $\rho$ using FIG score. In addition, the discussion about the intuition behind the forward and backward CT is helpful. Also it is good to see the discussion about the limitations of the proposed cost function. Given these new additions to the paper, I increased my score from 5 to 6.

---

> > > ### Author Response · Authors · 2021-08-29
> > > **Thank you for your feedback!**
> > >
> > > We sincerely appreciate your taking our responses into consideration and providing additional feedback to us. We will carefully incorporate these new additions mentioned in our responses to further strengthen the paper.

---

> ### Author Response · Authors · 2021-08-29
> **Request for feedback**
>
> Dear Reviewer aP1u,
>
> Thank you again for your detailed feedback. We hope our "Response to Two Common Questions" and the response shown below are sufficient to answer all your questions. We will be glad to provide further clarifications and answer any additional questions you may have.

---

### Official Review · Reviewer_ausY · 2021-08-01

**Rating:** 6
**Confidence:** 3

**Summary:**

This paper proposes to use conditional transport (CT) to measure the difference between two probability distributions. By combining both forward CT and backward CT, the proposed measure shows promising results for generative adversarial model training.


**Limitations And Societal Impact:**

No obvious potential negative societal impact.

**Main Review:**

After the discussion phase, I increase my overall score from 5 to 6. Thank the authors for their engagement.

--
Overall, this paper is well-written and the empirical results seem promising. However, I have two concerns:

1) Does the advantage of this new measure come from combining forward and backward CTs? If so, can we look at the results while changing the value of $\rho$?
2) How does the choice of cost function c(x,y) impact the performance of this measure?  Is Euclidean distance used for the all the experiments in this paper?


**Time Spent Reviewing:**

1

---

> ### Author Response · Authors · 2021-08-10
> **Response to Reviewer ausY**
>
> Thank you for your insightful questions. Below we provide a point-to-point response.
>
> > *Q1. Does the advantage of this new measure come from combining forward and backward CTs? If so, can we look at the results while changing the value of $\rho$?*
>
> A key advantage of combining forward and backward CTs is to help balance the mode-covering and mode-seeking behavior of the deep generative model trained under the bi-directional CT cost. Following your suggestion, we have provided example results under five different $\rho$ in “Response to Two Common Questions”. We also note the results in Fig.2 (a) on toy data and Table5 + Fig.13 (in appendix) on image data contain the results with different values of $\rho$.
>
> -----
>
> > *Q2. How does the choice of cost function c(x,y) impact the performance of this measure? Is Euclidean distance used for the all the experiments in this paper?*
>
> We mainly consider either Euclidean distance or cosine distance as the cost function.  When the data are in a low-dimensional space, where the Euclidean well reflects the difference between data points, we directly use Euclidean distance; when the data/encoder features are in a high-dimensional space, where the Euclidean distance often poorly reflects the difference between data points, we use the cosine distance to calculate the cost. While the Euclidean distance can also be a good choice to measure the difference between two high-dimensional data points in the encoder feature space, we find that using the cosine distance in this setting often leads to slightly/moderately better performance. The Table below provides a comparison of CT under two different cost functions on CIFAR-10.
>
>
> **Table. FID of generation on CIFAR-10, trained with different cost functions**
>
> | Cost function | \|\|x-y\|\|^2 | \|\|x-y\|\|^2 | 1-cos(x,y) | 1-cos(x,y) |
> |---------------|---------------|---------------|------------|------------|
> | Backbone      | DCGAN         | SNGAN        | DCGAN      | SNGAN      |
> | FID           | 27.1          | 18.0          | 22.1       | 17.2       |

---

> ### Author Response · Authors · 2021-08-29
> **Request for feedback**
>
> Dear Reviewer ausY,
>
> Thank you again for pointing out two concerns, which we have carefully addressed in our rebuttal. Would you please let us know whether our response is satisfactory? We will be glad to provide further clarifications and answer any additional questions you may have.

---

### Author Response · Authors · 2021-08-10
**Response to Two Common Questions**

We thank all reviewers for their valuable feedback. We first address two common questions.

> *Question 1: How the value of $\rho$ is selected and whether it needs to be tuned.*

- Unless specified otherwise, we chose $\rho=0.5$ as the default setting to balance between mode-covering and mode-seeking.

- While we have achieved good performance by simply fixing $\rho=0.5$, we agree tuning $\rho$ could lead to improved performance in terms of certain metrics, such as the FID score.
   - To verify this point, we have performed an additional ablation study on CIFAR-10 dataset with both the CT + DCGAN backbone and CT + SNGAN backbone.
   - The results, as shown in Table R1 below, suggest that the FID score could be further improved by choosing $0<\rho<1$ over fixing $\rho=0.5$.

  - As how to tune $\rho$ according to the dataset or downstream task is not the focus of this paper, we leave it for future study.


**Table R1. FID of generation results on CIFAR-10, trained with different $\rho$**

| $\rho$ |   0   |  0.25 |  0.5 |  0.75 | 1     |
|:---:|:-----:|:-----:|:----:|:-----:|-------|
| CT-DCGAN| 72.1 | **21.4** | 22.1 | 22.1 | 25.1 |
| CT-SNGAN| 33.2 | **17.2** | **17.2** | 17.5 | 23.2 |

> *Question 2: How to better understand mode-seeking and model-covering and formalize their definitions.*

- Suppose $p_X$ is the data distribution and $p_Y$ is the model distribution and they share the same support such that their KL divergence can be defined (note neither forward nor backward CT is subject to this constraint).

    - When using the forward KL divergence $KL(p_X||p_Y)=\mathbb{E}_{x\sim p_X(x)}[\ln \frac{p_X(x)}{p_Y(x)}]$ as the loss to optimize $p_Y$, we will need to ensure $p_Y(x)>0$ whenever $p_X(x)>0$, leading to a mode-covering behavior.

         - The forward CT cost (X->Y) shares similarity with the forward KL divergence on when it can be minimized: when $p_X(x)>0$, we need to make $\int c(x,y)\pi_Y(y|x)dy$ small to drive down the forward CT cost towards zero, which will not be possible if $p_Y(y)=0$ for the $y$’s that are close to $x$.
    - By contrast, when using the reverse (backward) KL divergence $KL(p_Y||p_X)=\mathbb{E}_{x\sim p_Y(x)}[\ln \frac{p_Y(x)}{p_X(x)}]$ as the loss to optimize $p_Y$, it is totally fine for $p_Y(x)=0$ when $p_X(x)>0$ and it is better to ask $p_Y$ to fit just some portion of $p_X$, leading to a zero-forcing and mode-seeking behavior.
         - The backward CT cost (X<-Y) shares similarity with the reverse KL divergence on when it can be minimized: when $p_Y(y)>0$, we need to make $\int c(x,y)\pi_X(x|y)dx$ small to drive down the backward CT cost towards zero, which can be achieved by only fitting some high density region of $p_X$.
 - Given these connections, we now introduce $$D(X,Y)= KL(p_X||p_Y)-KL(p_Y||p_X)$$ as a formal way to quantify the mode-seeking and mode-covering behavior of $p_Y$ with respect to $p_X$, with $D(X,Y)>0$ implying mode seeking and with $D(X,Y)<0$ implying mode covering.


 - For the specific toy example in Section 2.1, since the true data distribution is $\mathcal N(0,1)$ and the generator is $\mathcal N(0,e^{\theta}) $, we will observe the generator to exhibit mode-covering behavior when $\theta>0$, which is found to be the case when only the forward CT cost is being optimized, and mode-seeking behavior when $\theta<0$, which is found to the case when only the backward CT cost is being optimized.
     - The forward KL is: $KL[\mathcal N(0,1)|| \mathcal N(0, e^\theta)] = \frac{1}{2}( \theta + e^{-\theta} - 1)$;
     - The reverse KL is: $KL[ \mathcal N(0, e^\theta) || \mathcal N(0,1)] = \frac{1}{2} ( e^{\theta}  - \theta - 1)$;
     - The difference is  $D(X,Y)=  KL[\mathcal N(0,1)|| \mathcal N(0, e^\theta)] - KL[ \mathcal N(0, e^\theta) || \mathcal N(0,1)] = \theta - sinh(\theta)$
      -  One can show that $D(\mathcal N(0,1),  \mathcal N(0, e^\theta)) > 0$ if $\theta < 0$ and $D(\mathcal N(0,1),  \mathcal N(0, e^\theta)) <0$ if $\theta > 0$




- As an empirical verification, we calculate $D(X,Y)$, with the 1d-GMM empirical fitting results shown in the upper panel of Fig. 2(a). Here we take 200 grids in the range [-10,10] to approximate the empirical distribution $\hat p_X$ and $\hat p_Y$. The corresponding results,  shown in the table below, further verify the model-seeking behavior of the backward CT (i.e., $\rho=0$) and the model-covering behavior of the forward CT (i.e., $\rho=1$).


| $\rho$                         | 0     | 0.25  | 0.5   | 0.75 | 1     |
|--------------------------------|-------|-------|-------|------|-------|
| $KL[\hat p_X \|\| \hat p_Y]$    |  10.52 |    0.94 |    0.12  |  0.23    | 1.97  |
| $KL[\hat p_Y \|\| \hat p_X]$    | 3.52  |      0.80      |   0.19  |  0.40  | 3.57  |
| $\hat D(X,Y)$ | 7.00  | 0.14  | -0.07 | -0.17 |  -1.60 |

---

### Decision · Program_Chairs · 2021-09-27

**Decision:**

Accept (Poster)

**Comment:**

Three reviewers recommend an accept, one reviewer indicates a reject. The general sentiment is that the presented approach for comparing probability distributions with forward and backward CT is novel, and that the empirical results for generative adversarial model training are promising. However, there were some concerns as to the lack of theoretical results and inadequate discussion about the limitations of the proposed method. A revised paper should incorporate the revisions and clarifications brought up in the rebuttal.